# Robust dynamic community detection with applications to human brain functional networks

L.-E. Martinet[1,5], M. A. Kramer [2,3,5], W. Viles[2], L. N. Perkins [4], E. Spencer [4], C. J. Chu[1], S. S. Cash[1] & E. D. Kolaczyk [2✉]

While current technology permits inference of dynamic brain networks over long time periods at high temporal resolution, the detailed structure of dynamic network communities during human seizures remains poorly understood. We introduce a new methodology that addresses critical aspects unique to the analysis of dynamic functional networks inferred from noisy data. We propose a dynamic plex percolation method (DPPM) that is robust to edge noise, and yields well-defined spatiotemporal communities that span forward and backwards in time. We show in simulation that DPPM outperforms existing methods in accurately capturing certain stereotypical dynamic community behaviors in noisy situations. We then illustrate the ability of this method to track dynamic community organization during human seizures, using invasive brain voltage recordings at seizure onset. We conjecture that application of this method will yield new targets for surgical treatment of epilepsy, and more generally could provide new insights in other network neuroscience applications.

[1] Department of Neurology, Massachusetts General Hospital, Boston, MA 02114, USA. [2] Department of Mathematics and Statistics, Boston University, Boston, MA 02215, USA. [3] Center for Systems Neuroscience, Boston University, Boston, MA 02215, USA. [4] Graduate Program in Neuroscience, Boston University, Boston, MA 02215, USA. [5] These authors contributed equally: L.-E. Martinet, M. A. Kramer. ✉email: kolaczyk@bu.edu

The brain functions (and dysfunctions) through interactions spanning spatial scales, from the single neuron to the entire nervous system, and temporal scales, from millisecond action potentials to decades of development[1]. Modern neuroimaging, combined with sophisticated data analysis tools, has expanded analysis of brain activity from individual brain components to networks of interacting brain regions. Understanding these brain networks (e.g., the connectome[2], dynome[3], and chronnectome[1]), and the big data they entail, remains a fundamental challenge of modern neuroscience[4], with the potential for significant impacts to human health and disease[5,6].

Initial research efforts have focused on characterizing properties of isolated brain networks using static graph metrics, resulting in candidate network features important to brain function, including hubs[7,8], rich-clubs[9,10], and small-worldness[11]. However, brain networks are not static; instead, the patterns of brain connections change in time, and emerging research suggests these network dynamics are critical to brain function[12] and dysfunction[13–15]. Although many tools have emerged to characterize network dynamics[16,17], the most common is dynamic community detection (i.e., tracking how a group of nodes that share increased connections changes in time). These methods typically apply an algorithm developed for static graphs to define candidate communities at a fixed time, and then define time-varying communities by linking consecutive static communities. For example, the clique percolation method (CPM) defines communities in static time slices by the extent to which a clique (i.e., a fully connected subnetwork) can be walked over the graph, and then communities at successive time points are linked using a rule based on overlap of vertex subsets[18]. In neuroscience, the most popular method to detect communities in temporal networks is the multilayer modularity method (MMM), which extends the standard modularity maximization framework to uncover communities across time[19,20]. Application of MMM has provided new insights into many areas of brain function, including learning[21,22], aging[23,24], language[25], and cognition[26,27].

Despite their widespread application in neuroscience, two key challenges face existing dynamic community detection methods. First, existing methods generally are unable to account for the edge noise that is inherently present in functional networks inferred from noisy brain voltage data[28]. Specifically type-II errors, or false negatives, are problematic given that type-I errors typically are controlled as part of the network inference process. Second, existing approaches generally lack a definition of communities that is explicit and interpretable (e.g., in terms of basic network motifs[29]) both within and across slices of time. For example, MMM employs an optimization criterion[19,30], yielding an implicit notion of community (albeit computationally tractable and mathematically elegant).

Here, we develop and apply a new dynamic community detection method to address these challenges. The proposed method extracts dynamic communities based on the explicit notion of time-evolving aggregations of smaller motifs, which have been proposed as building blocks characteristic of different types of networks[31,32]. The new method of dynamic community detection—the dynamic plex percolation method (DPPM)—connects static communities within a time slice to aggregations of a variety of common motifs in a natural and flexible manner, defines dynamic communities across time through an explicit notion of temporal progression of these motifs, and is demonstrably robust to edge noise. We show in simulation that DPPM outperforms existing methods in four stereotypical community evolution scenarios. We then apply DPPM to a dataset derived from invasive brain voltage recordings made from human subjects during seizures. Our analysis demonstrates a large dynamic community that rapidly grows from a spatially localized region at seizure onset, suggesting the potential for targeted therapeutic intervention.

## Results

**The DPPM**. The DPPM operates on dynamic networks, which may be inferred from noisy time series data. Before illustrating the utility of this method in simulation and examples of observational data, we first briefly describe and motivate the proposed dynamic community detection procedure.

Let $G = (V,E)$ be a graph with vertex set $V$ and edge set $E$, and let $\{G_t\}_{t=1}^N$ denote a sequence of $N$ graphs indexed by time $t$, which we assume to share a common vertex set $V$. Our goal is to identify dynamic communities in such a sequence of graphs. We define a community explicitly as subsets of nodes—within and across time—that are reachable by small template subgraphs that are walked within and across temporally adjacent graphs $G_t$. The result of this definition is that our communities may be conceptualized as an evolving series of tubes, which represent cohesive aspects of the dynamic networks evolving in time.

In walking, movement necessarily must be from one copy of the template subgraph to another in such a way that the two differ by at most one vertex. Our choice of template subgraphs are plexes. A $k$-plex of size $m$ is a vertex-induced subgraph $S$ of $m$ vertices from a graph $G$ with the property that the degree of $v$ in the subgraph is at least $m-k$ for all $v \in S$ and $m > k$. In other words, the order $k$ refers to the maximum number of missing neighbors. Movement across time is facilitated by artificially connecting vertex pairs across $G_t$ and $G_{t+1}$, for every $t$, in a manner conducive to walking plexes. Specifically, (1) each vertex $v \in G_t$ is connected by an edge to its mature self in $G_{t+1}$, and (2) if an edge $\{v_1, v_2\} \in G_t$ exists again in $G_{t+1}$ then the vertices $v_1 \in G_t$ and $v_2 \in G_{t+1}$ are connected by an edge, and likewise $v_2 \in G_t$ and $v_1 \in G_{t+1}$. The result of these two steps is to create a proper bridge for plex walking (Fig. 1a). This enhanced version of the sequence $\{G_t\}$ is the infrastructure upon which the plexes walk in space and time, and thus dynamic communities are well defined. Note that this enhanced graph sequence is independent of the choice of both plex size and order. For a detailed description of the algorithm, including pseudo-code and comments on implementation, please see "Methods".

The central role played by plexes in the DPPM framework derives from the fact that plexes are network elements consistent with motifs, the building blocks of larger network structures[31,32] (Supplementary Table 1). Our dynamic communities, explicitly defined as aggregations of such building blocks, are thus consistent in spirit with notions of network (sub)structure in network science. In this sense, DPPM is an extension of the CPM[18]. DPPM differs from CPM, however, in that the latter uses the more rigid notion of a clique (a fully connected subgraph, $k = 1$) as a template subgraph. Importantly, replacing cliques by the more flexible notion of plexes leads to robustness against edge noise (in particular type-II errors, i.e., false negatives), which is common in functional networks inferred from multisensor brain recordings.

Community detection is an area that has seen extensive development, and the literature on dynamic community detection is already nontrivial[33]. Nevertheless, our work below shows that substantial improvement on current state-of-the-art is still possible where specific questions relating to notions like coalescence and fragmentation of dynamic communities is concerned (illustration in Fig. 1b). Increasingly, evidence suggests that such notions are likely central to better understanding the evolution of phenomena like the seizures motivating our work[34–36].

As representative comparisons, we focus on two other methods of dynamic community detection popular in the analysis of

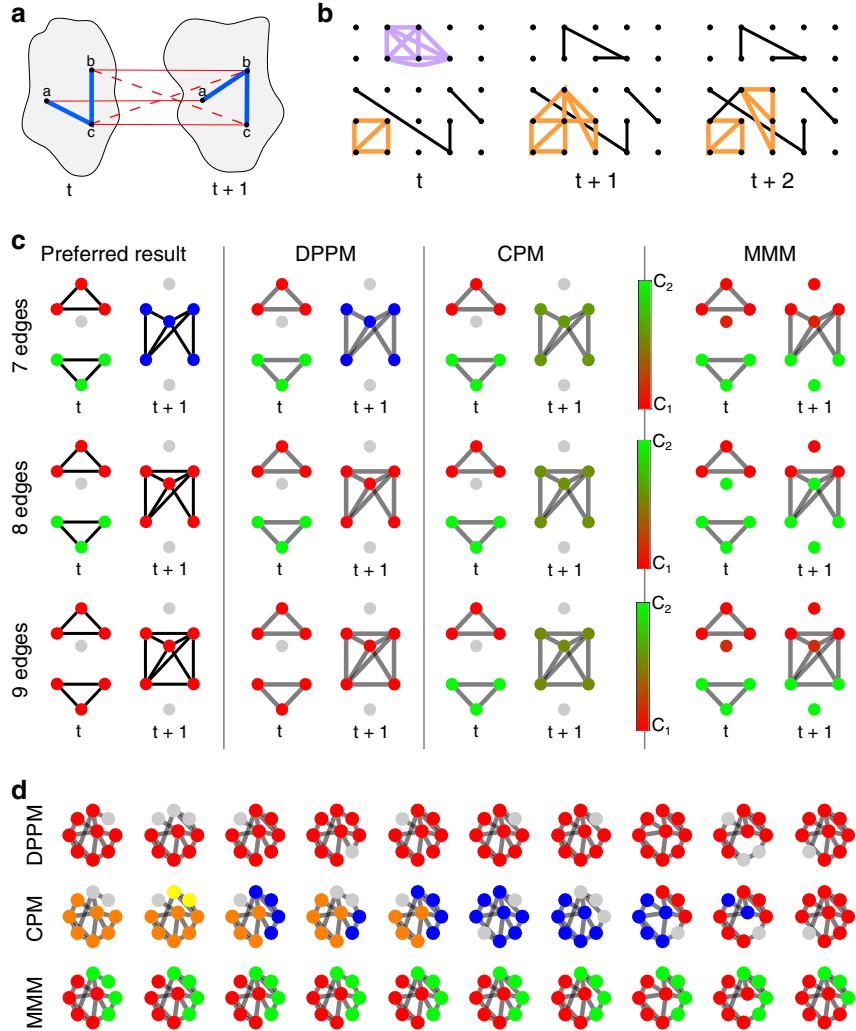

**Fig. 1 Illustration of DPPM principles and effectiveness. a** Schematic of the bridges used to walk plexes within dynamic communities across time. Blue edges represent inferred connectivity, while red edges connecting the same (solid) or adjacent (dotted) vertices facilitate movement. **b** Illustrative example showing how a simple community (in orange) is tracked by DPPM across time in a manner allowing for both coalescence and fragmentation. This community at first grows at $t + 1$, and then fragments at $t + 2$. Another community present at $t$ (purple) perishes at $t + 1$. **c** Comparison of DPPM (with 2-plex of size $m = 3$, 2nd column), CPM (with 1-plex of size $m = 3$, 3rd column), and MMM (with $\gamma = 1$, $\omega = 1$, 4th column) in determining communities across two adjacent time points $t$ and $t + 1$. The connected component at time $t + 1$ shares increasingly more edges components at time $t$ from the top to the bottom row of plots. In the preferred results, the dynamic communities (1st column) depend on the number of edges shared from time $t$ to $t + 1$. Whereas DPPM treats these as three distinct scenarios, neither CPM nor (effectively) MMM distinguish the three cases. In CPM and MMM, the colorbar indicates the proportion of community membership over $n = 100$ repetitions of community detection. **d** Example dynamic community tracking in the presence of missing edges. Dynamic community membership for ten example sequential time index networks computed using DPPM (with 2-plex of size $m = 4$), CPM (with 1-plex of size $m = 3$), and MMM (with $\gamma = 1$, $\omega = 1$). While DPPM detects a single dynamic community in time, the other two methods do not.

functional connectivity networks: CPM and MMM[19,30]. DPPM differs from CPM not only in its use of plexes rather than cliques, but also in the manner that those are used to define dynamic communities. Specifically, CPM only walks the clique within time slices $G_t$ to identify static communities, rather than both within and across time slices (as in DPPM), using instead a more ad hoc rule of overlap among static communities across adjacent times to form dynamic communities. MMM, in contrast, differs from both DPPM and CPM in its core mechanism, adopting an implicit notion of communities defined via optimization of a cost criterion like modularity.

To illustrate the impact of these differences, we consider three simple examples of dynamic community tracking between two sequential networks (Fig. 1c). In the first example, two connected components at time $t$ evolve to a single connected component at time $t + 1$ (Fig. 1c, seven edges). While DPPM detects three separate communities, both CPM and MMM detect only two communities. We note that, unlike the deterministic result of DPPM, the community label at time $t + 1$ for CPM and MMM is not unique. In this example, the community label for CPM at time $t + 1$ is chosen arbitrarily (i.e., by a fair coin flip) from the existing community labels at time $t$[37]. In the second example, we add an additional edge to the single connected component at time $t + 1$ (between the upper two nodes, Fig. 1c, eight edges). In this case, DPPM tracks a dynamic community from time $t$ to $t + 1$ (red in Fig. 1c, eight edges). While the DPPM result is quite different in this example, the additional edge has little effect on the dynamic communities detected with CPM and MMM. Addition of another edge (between the lower two nodes at $t + 1$) again changes the DPPM result (Fig. 1c, nine edges); in this

case, all components at times $t$ and $t+1$ establish a single community. This occurs because a plex, beginning in the upper triangle at time $t$, can propagate to the single component at time $t+1$, and then back to the lower triangle at time $t$. Again, the addition of an edge has little effect on the dynamic communities detected with CPM and MMM. We note that the choice of MMM parameters $\gamma = 1$, $\omega = 1$ serves as a representative example; other parameter choices perform similarly (see Supplementary Fig. 1). We conclude that unlike the latter two methods, DPPM distinguishes between the three subtly distinct basic scenarios in an explicit and deterministic way.

In addition, we demonstrate the robustness of DPPM to noise. To do so, we begin with a nine-node template network possessing eight edges. We then replicate this network 100 times to create a time-indexed network, such that at each time we remove two randomly chosen edges. We expect that a dynamic community tracking procedure robust to noise (i.e., type-II errors or missing edges) would track a single community in time. We find that DPPM succeeds and successfully tracks a single dynamic community in time (Fig. 1d, top row), whereas CPM fractures into different transient dynamic community detections (Fig. 1d, middle row). While MMM also detects a consistent structure in time, it consists of two dynamics communities, based on the configuration of the eight edges in the template network (Fig. 1d, bottom row). We conclude that edge noise dramatically impacts CPM, while both DPPM and MMM are robust to noise, but only DPPM detects a single community in time.

**DPPM accurately tracks dynamic communities**. To better characterize the performance of DPPM compared with existing dynamic community detection methods, we consider four categories of simulation. Each category simulates a representative dynamic community behavior, motivated by the notion that brain functional networks can expand, contract, split, and merge in response to changing task demands, internal states, and disease[18]. We show that, across these four categories, DPPM outperforms existing methods and successfully detects the functional network community dynamics.

We start with simulations of community expansion. The simulations begin with a group of nodes that belong to the same community (Fig. 2a, b). As time evolves, a second group of nodes joins the community established by the first group (Fig. 2a, b). At a later time, a third group joins the expanding dynamic community. In this way, the community expands to encompass larger groups of nodes as time evolves. We note that all nodes that join the expanding group share a common community label, while nodes outside of this expanding group are assigned a random community labels (see "Methods").

Application of the three dynamic community detection methods yields distinctly different results. While in this example CPM fails to capture the community expansion, both DPPM and MMM succeed (Fig. 2c). We note that here, and in the simulations that follow, we fix the detection method parameters to show representative examples; see Supplementary Fig. 2 for examples of each community detection method applied with different parameter settings.

In the second category of simulation, we consider dynamic community contraction, which can intuitively be understood as dynamic community expansion with time reversed. For these simulations, we begin with a large group of nodes that share a common community label (see ground truth on Fig. 3a, left panel). As time evolves, we remove nodes from this community. By doing so, we expect to reduce the number of nodes that participate in the dynamic community, until eventually all nodes are removed (and each node assigned a random community label,

see "Methods"). In this illustrative example, only DPPM successfully detects the dynamic community contraction with high sensitivity and specificity (Fig. 3a). Different choices of module size in CPM, or structural and temporal resolution parameters in MMM, tend to detect the dynamic network contraction with low sensitivity, low specificity, or both (Fig. 4b, e; see Supplementary Fig. 3 for examples of each community detection method applied with different parameter settings).

For the third simulation category, we consider the scenario of one community splitting into two. To simulate this scenario, we begin with a single group (see Group A on Fig. 3b, left panel) of nodes that share a common community label (see ground truth in Fig. 3b, left panel). After an interval of time, two new groups of nodes appear (Groups B and C, Fig. 3b). Each node in these groups shares a common community label with Group A. All three groups remain active for an interval of time, establishing a dynamic community consisting of nodes from all three groups. Then, Group A leaves the common community, while Groups B and C remain in the common community. We expect to detect a single dynamic community splitting from the collection of all nodes in Groups A, B, and C, to the collection of nodes only in Groups B and C (i.e., the blue ground truth community in Fig. 3b, left panel).

We find that all three methods detect some aspects of the dynamic community splitting (Figs. 3b and 4c, e; see Supplementary Fig. 4 for examples of each community detection method applied with different parameter settings). However, only DPPM detects the dynamic community splitting with high sensitivity and specificity.

Finally, we consider the converse of splitting: dynamic community merging. In this scenario, we begin with two groups (Groups B and C) whose nodes share a common community label (see ground truth in Fig. 3c, left panel). After an initial interval of time, a third node group (Group A) joins the common community. Finally, we remove nodes only in Group B and Group C (Fig. 3c). After these removals, only the Group A nodes share a common community label. We expect to detect a single dynamic community that begins with nodes in Groups B and C, then adds nodes from Group A, and finally consists only of Group A nodes. We again find that, although all three methods capture some features of the dynamic community merging, only DPPM performs with high sensitivity and specificity (Figs. 3c and 4d, e; see Supplementary Figs. 5 and 6 for examples of each community detection method applied with different parameter settings).

Based on the results from these four simulation categories (summarized in Fig. 4), we conclude that only DPPM detects the dynamic community behavior with high sensitivity and high specificity in all scenarios considered. We find that no fixed parameter setting for MMM performs with both high sensitivity and high specificity across all simulation scenarios. While DPPM and CPM perform with similar specificity across parameter settings, DPPM (with $m=4$, $k=2$; $m=4$, $k=3$; or $m=5$, $k=3$) perform with higher sensitivity than CPM.

**Expansion of a dynamic community at human seizure onset**. To illustrate the performance of DPPM on clinical data, we consider an application to invasive electrocorticogram (ECOG) recordings from an $8 \times 8$ electrode grid placed directly on the cortical surface of a human patient undergoing resective surgery for epilepsy (see "Methods"). From these data, we infer dynamic functional networks (see "Methods") that begin 100 s before seizure onset, and examine the dynamic communities that emerges at seizure onset in four seizures. We choose to examine seizure onset, where a rapid increase in functional connectivity often

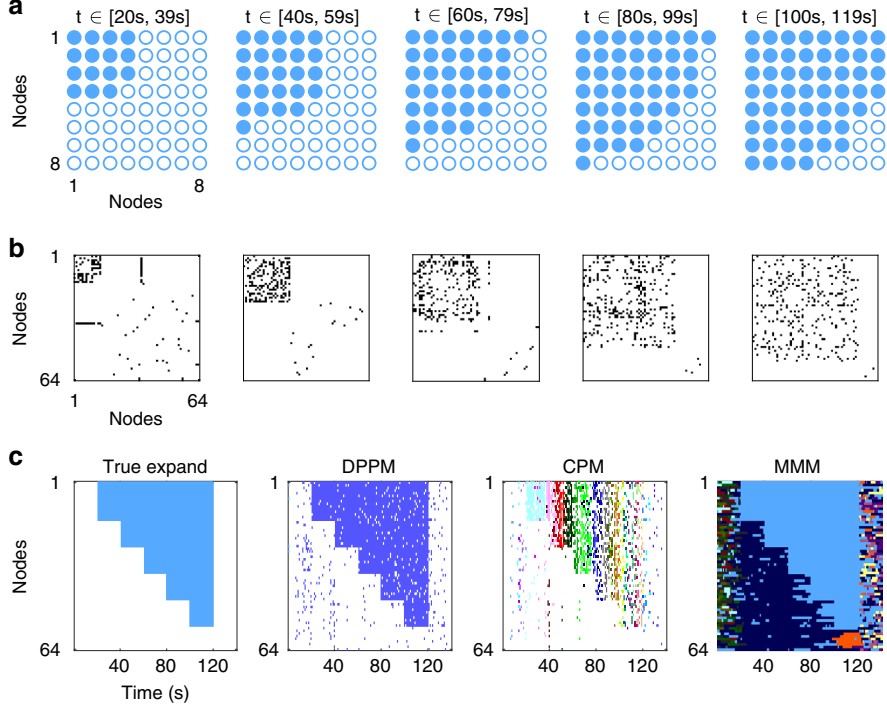

**Fig. 2 In an example of dynamic community expansion, DPPM outperforms two existing methods.** Illustration of community expansion in nodes (**a**) and in edges (**b**). **a** Two-dimensional representation of the nodes on an 8 × 8 grid at five time intervals. Color (blue) indicates when a node becomes recruited to the largest community. **b** Adjacency matrices for the simulated network of 64 nodes at the same five time intervals. Color (black) indicates an edge between a node pair. **c** Dynamic community detection results for each method, from left to right: true expansion, DPPM with parameters $m = 4$ and $k = 2$, CPM with parameter $m = 4$, and MMM with parameters $\gamma = 0.1$, $\omega = 0.1$; see Supplementary Figs. 2 and 6a for results with different parameter choices, and see Fig. 4a, e for average results over 100 realizations of the simulation. Color indicates community membership. The largest community detected by DPPM is most consistent with the true expansion.

occurs[38,39]. Before seizure onset, a large number of small dynamic communities transiently appear ($N = 1768$ communities detected for $n = 4$ seizures, example in Fig. 5a). The vast majority of these communities are short-lived (median lifespan 1 s, 95% of lifespans <6.55 s, gray histogram in Fig. 5c) and spatially limited (median maximum size 4 nodes, 95% of maximum community sizes <13 nodes, Fig. 5d). After seizure onset, a large dynamic community appears and persists for tens of seconds before contracting and disappearing (example in Fig. 5a, middle panel, the red community); we label this community the seizure onset community. During expansion, the seizure onset community rapidly recruits nodes (e.g., red curve and red community in Fig. 5a, b) until almost all nodes are recruited (median maximum size 61 nodes for $n = 4$ seizures, see arrows in Fig. 5d) and stays active for significantly longer than the pre-seizure communities (median lifespan 143.50 s, range [55 s, 239.5 s], $n = 4$ seizures, arrows in Fig. 5c).

Computing node loyalty (i.e., how often a node is part of a community), we find that, before seizure onset, two brain regions are more likely to participate in the multiple communities observed (Fig. 5e). During seizures, the spatial pattern of node loyalty changes to become broader (Fig. 5f); more nodes participate longer in the seizure onset community than in the pre-seizure communities. We also observe that the maximum node loyalty emerges from a spatially localized brain region, centered on a region of high pre-seizure node loyalty (Fig. 5e, f, black circle), perhaps consistent with an ictogenic process emerging before clinical seizure onset.

To characterize how nodes join the seizure onset community, we calculate the node recruitment order (i.e., an ordering of the nodes from first to join the seizure onset community to last to

join). Visual inspection of the median maps of node recruitment order (Fig. 5g) across the patient's four seizures suggests recruitment occurs in a spatially organized pattern; neighboring nodes tend to be sequentially recruited in the seizure onset community. We note that the spatial pattern of this node recruitment is consistent with the spatial pattern of recruitment into large amplitude oscillations after seizure onset for this patient[40] (Fig. 5h). Perhaps surprisingly, analysis focusing on two different aspects of seizure dynamics (dynamic functional connectivity here versus the dynamics of signal power in[40]) produce consistent spatial maps (compare Fig. 5g, h). Moreover, we note that the time scale over which these spatial maps appear differ by an order of magnitude; recruitment to the seizure onset community occurs within seconds, while recruitment to the large amplitude oscillations characteristic of seizure occurs over tens of seconds. Yet, despite this temporal difference, the spatial maps appear consistent.

To explore further these initial observations we repeat the analysis for a population of patients and seizures (12 patients, 38 seizures). We find, consistent with the example patient and seizures, that: (1) the largest community size increases during seizure compared with pre-seizure (Fig. 5i), and (2) the lifespan of the largest community is longer during seizure compared with pre-seizure (Fig. 5j). We then explored the hypothesis that patients with a worse surgical outcome would have more fractured, longer-lasting dynamic communities during seizure, which are less susceptible to a targeted intervention (in this case, resective surgery). We find that both the number of communities (Fig. 5k) and the duration of the longest community (Fig. 5l) are higher in patients with worse surgical outcomes ($p = 0.025$, t-statistic $= -2.35$, 95% confidence interval for the difference in

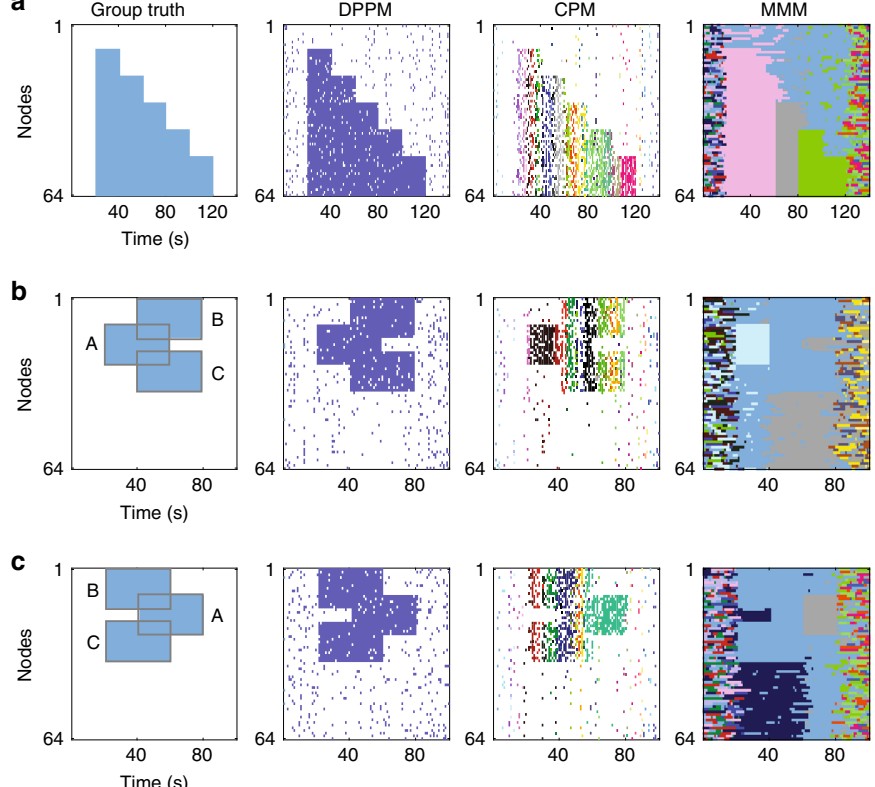

**Fig. 3 In three additional examples of dynamic community evolution, DPPM outperforms two existing methods.** Community detection for each method in the case of **a** community contraction, **b** community splitting, and **c** community merging. From left to right in each subfigure: true community evolution, DPPM with parameters $m = 4$ and $k = 2$, CPM with parameter $m = 4$, and MMM with parameters $\gamma = 0.1$, $\omega = 0.1$; see Fig. 4 and Supplementary Figs. 3–6 for results with different parameter choices. Color indicates community membership. In all cases, the largest community detected by DPPM is most consistent with the true community evolution.

population means [−0.59, −0.04]; and $p = 0.0016$, t-statistic = −3.41, 95% confidence interval for the difference in population means [−0.46, −0.11]; respectively, two-tailed two-sample $t$-test, $n = 27$ from nine patients with low Engel score, and $n = 11$ from three patients with high Engel score, degrees of freedom = 36).

We conclude from these observations that application of DPPM to ECOG data recorded from patients with epilepsy provides new insights into the expansion of a large dynamic community at seizure onset. We propose that larger communities of longer duration emerge during seizure, and that patients with fewer, shorter duration communities during seizure have improved surgical outcomes.

## Discussion

In this manuscript, we introduced the DPPM to track dynamic communities that evolve in time. We designed this method specifically to address the challenges introduced when inferring dynamic functional networks from neural time series data. The resulting method differs from existing dynamic community tracking procedures in two ways. First, we extract communities based on the explicit notion of time-evolving aggregations of smaller motifs, rather than through optimization of a cost criterion. Second, we account for edge noise—a factor in any set of functional networks inferred from time series data. We showed in simulations that DPPM outperforms two existing methods in representative dynamic network scenarios motivated by the intuitive notions of community expansion, contraction, merging, and splitting. We then applied DPPM to examples of time-indexed functional networks inferred at human seizure onset. We showed preliminary evidence of rapid, dynamic community

expansion at seizure onset from an initial set of nodes, and that community structure during seizure correlates with surgical outcome.

The results presented here may appear to challenge previous applications of dynamic community evolution procedures to human brain activity[21–23,25–27,30,41–44]. However, one should not dismiss this prior work. We showed in simulation that MMM may still produce accurate results, and the ability to investigate a range of resolution parameter values may provide a more comprehensive view of a network's modular organization[45]. Different brain activity (e.g., slow hemodynamic responses versus fast electrophysiological changes) may results in dynamic functional networks more compatible with accurate inference over a broad range of resolution parameters. While DPPM only links nodes at neighboring time steps, MMM permits links between nodes across broader time intervals. Such links may support more accurate detection of communities in which nodes infrequently—but consistently—participate. We note that the communities identified by any method, while not necessarily optimal, could still facilitate fruitful exploration of functional brain networks. For example, the distinction between a large community, and two smaller communities that evolve similarly, may not have practical consequences to understanding brain function or dysfunction.

While DPPM performs well in simulations, and in an example application to in vivo data, we note three limitations. First, the computational time required to track dynamic networks with DPPM depends on the plex size, and hence is likely to be most practical for plexes of relatively small scale. We developed approximations to reduce computational time in two cases—(4,2) and (5,3)—which are consistent with the size of structural and

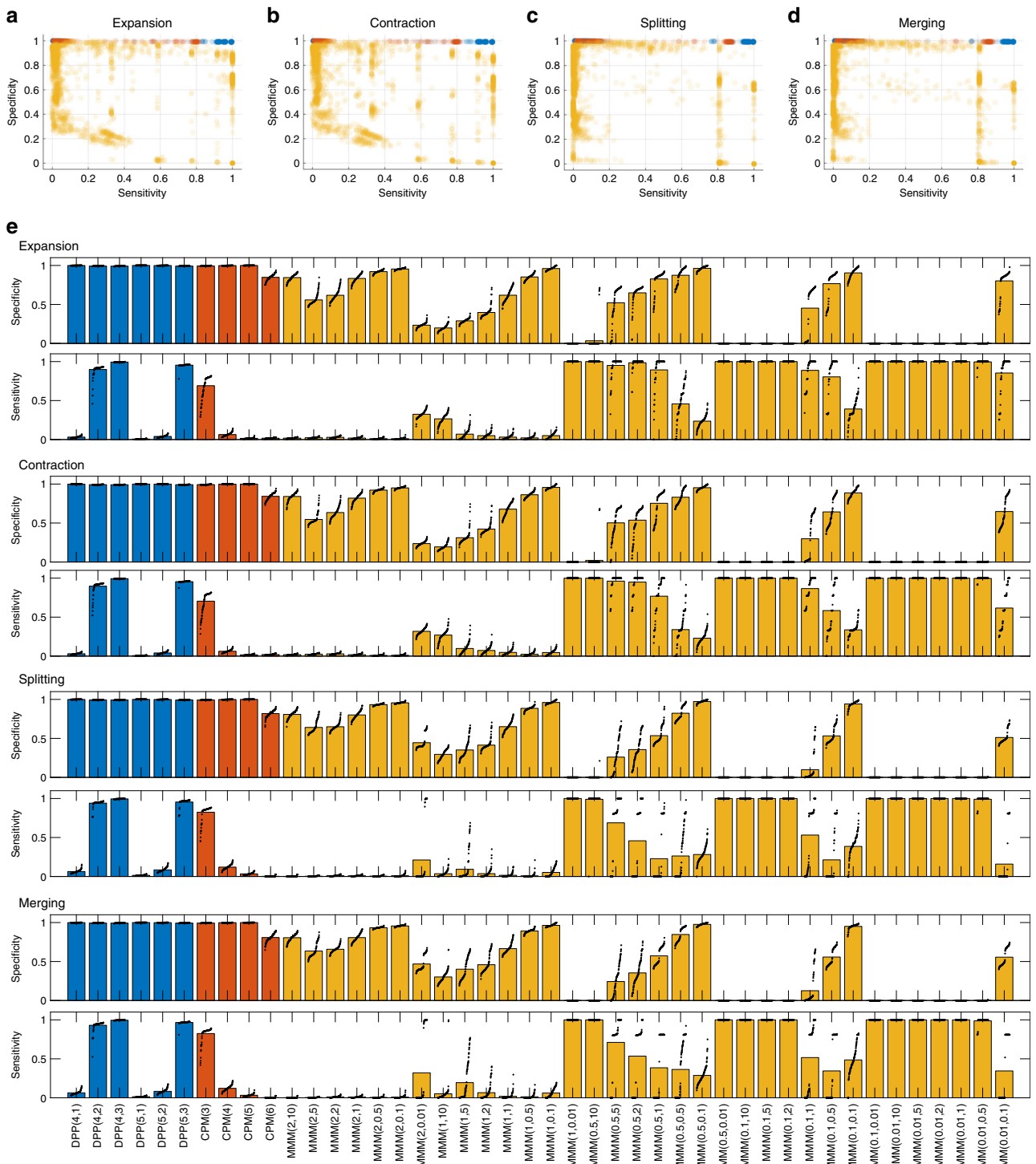

**Fig. 4 DPPM performs with higher sensitivity and specificity than two existing methods in the four simulation scenarios.** Specificity and sensitivity of DPPM (blue), CPM (red), and MMM (yellow) for $n = 100$ independent simulations with different noise instantiations of each dynamic community evolution scenario. **a** Community expansion, **b** Community contraction, **c** Community splitting, and **d** Community merging. Each circle indicates the result of one simulation with one parameter configuration (see "Methods"). **e** Summary results for each community tracking method applied to each simulation scenarios. Bars indicate the mean sensitivity and specificity for each parameter configuration of each method. Dots indicate the results of the $n = 100$ simulations for each simulation scenario and method.

functional motifs proposed as common network building blocks[31,32]. DPPM successfully aggregates these common motifs into communities. We note that the appropriate interpretation of network motifs in neuroscience remains a point of open discussion. While motifs have been proposed as network building blocks of the brain[46], motifs may instead represent a byproduct of local brain connectivity[47,48]. The connection of DPPM with motifs through the central role of plexes may facilitate more explicit study of this issue going forward, in the specific context of dynamic community detection. Applications to different systems,

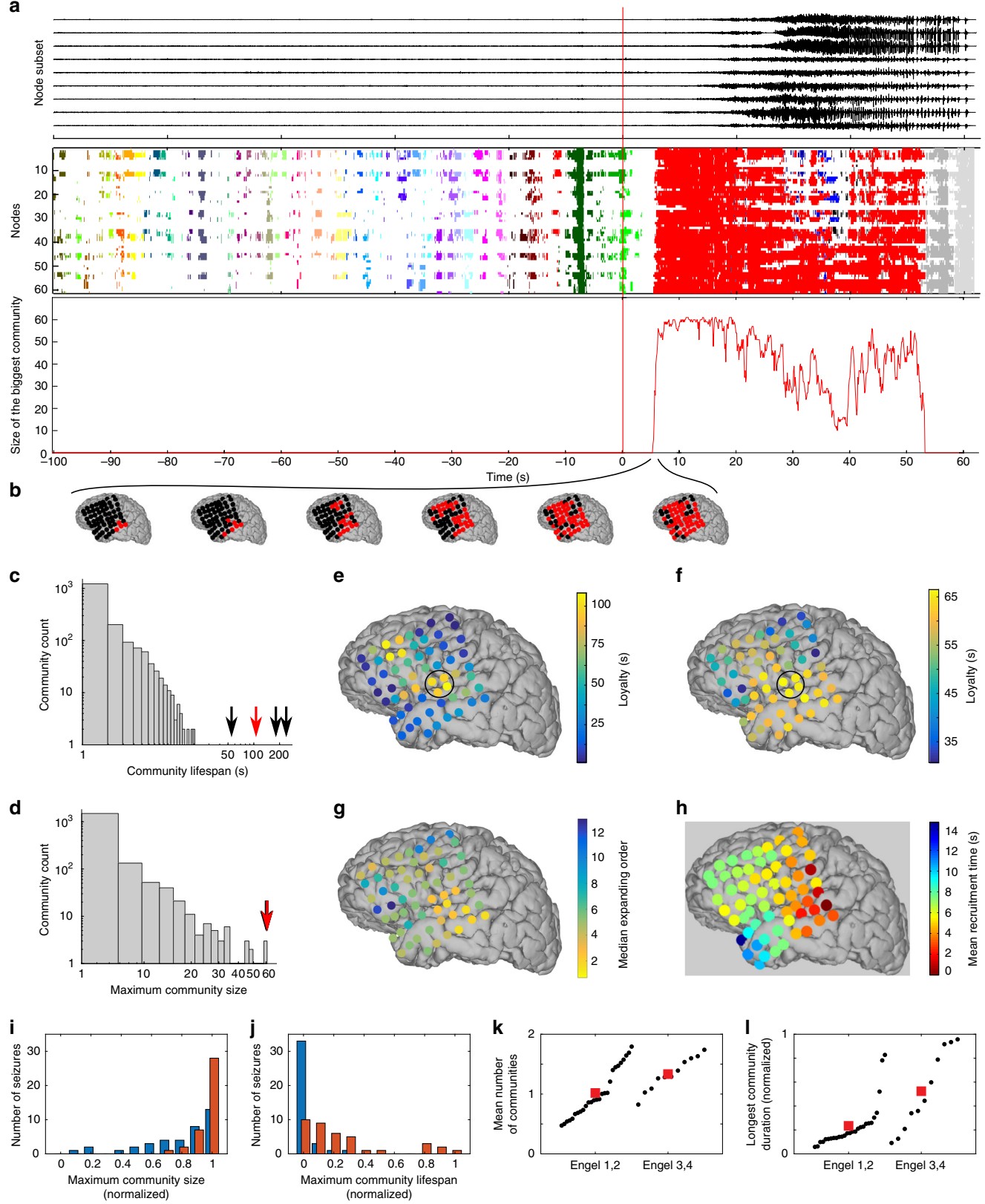

in which larger plexes are required, may require additional development. Along this line, maximal $k$-plex identification, which is a computationally costly algorithm at the core of DPPM, has received renewed interest in network sciences, which could lead to further speed improvements[49,50].

Second, DPPM may link dynamic communities which may be interpreted as separate based on information beyond the network connectivity. To illustrate this, we consider the hypothetical scenario in which two dynamic communities exist simultaneously. Within one community, a low frequency rhythm (e.g.,

**Fig. 5 Application of DPPM reveals new characteristics of dynamic communities before and after human seizure onset. a** Top: Voltage time series recorded at nine electrodes to illustrate pre-seizure and seizure voltage dynamics. Middle: Example recruitment of a large community at seizure onset. Before seizure onset ($t < 0$ s) small communities appear briefly; color indicates community membership. After seizure onset ($t > 0$ s) a large dynamic community appears (red) that persists for over 30 s. Bottom: Temporal evolution of the size of the seizure onset community (red). Nearly all nodes participate in the seizure onset community. **b** Example expansion of the seizure onset community. Each circle denotes an electrode on the $8 \times 8$ electrode grid, and red (black) indicates electrodes recruited (not yet recruited) into the dynamic community. Community lifespan (**c**) and maximum community size (**d**) for all pre-seizure communities (gray histograms) and the seizure onset community observed for each of the four seizures of this patient (four arrows, red indicates the community shown in (**a**)). Node loyalty averaged over the four seizures for (**e**) all pre-seizure communities and (**f**) the seizure onset community. In both panels, warm (cool) colors indicate nodes that participate in communities for longer (shorter) times. The black circles indicate a subset of electrodes that have high node loyalty before and during early seizure. **g** Median recruitment order to the seizure onset community for four seizures. Warm (cool) colors indicate electrodes recruited earlier (later) into the seizure onset community. **h** Mean recruitment time to large amplitude ictal oscillations observed for the same patient, as reported in ref. [65]. Warm (cool) colors indicate electrodes recruited earlier (later) into ictal spread. Histograms of the size (**i**) and the lifespan (**j**) of the maximum community during the pre-seizure (blue) and seizure (red) intervals from each patient and seizure. The maximal community tends to be larger and of longer duration during seizure. The mean number of communities (**k**) and the longest community duration (**l**) during seizure for patients with good (Engel 1,2) and poor (Engel 3,4) surgical outcomes. Each circle indicates an individual seizure, and the red square the population mean ($n = 27$ from nine patients with low Engel score, and $n = 11$ from three patients with high Engel score). Worse surgical outcomes exhibit more communities with longer maximal duration during seizure.

theta, 4–8 Hz) links the nodes, while in the other dynamic community, a higher frequency rhythm (e.g., beta, 12–20 Hz) links the nodes. Initially, because different frequency rhythms appear in each community, no links exist between the two communities; i.e., the communities are functionally disconnected. Then, at a later time, nodes in both communities transition to the same rhythm (e.g., alpha, 8–12 Hz), establishing functional connections between the nodes in both communities, and causing the two dynamic communities to merge. In this scenario, DPPM would identify a single dynamic community that includes all nodes at all time considered. That the two sets of nodes initially form separate functional communities—employing rhythms in different frequency bands—is not represented in the single dynamic community identified by DPPM. Analysis of node properties (e.g., the power spectrum) or selection of coupling measures targeting specific frequency bands would address this particular scenario.

Third, DPPM is designed for binary networks that are often inferred from multi-electrode brain recordings, but in neuroscience it is frequently desirable to analyze weighted networks. CPM has been extended to weighted networks[51], where in order to percolate cliques are now required to be of sufficient intensity (i.e., the geometric mean of the link weights in a clique[52] must exceed a given threshold). An extension of DPPM to weighted networks should be similarly feasible, although somewhat less immediate for two reasons. First, there can be a multiplicity of plexes among a given set of nodes, and thus there is flexibility in representation. One approach might be to adopt the recently-introduced notion of a maximal edge-weighted plex[53]. Second, it is necessary to equip the edges between the same nodes at different network time points with an appropriate notion of an edge weight. Depending on the manner of network construction, there may be multiple ways in which to do so. A full and careful exploration of these possibilities is beyond the scope of the present manuscript. We note that MMM handles weighted edges seamlessly and without modification.

While tools from network analysis serve an essential role in understanding multisensor recordings[54], significant challenges remain in the application of these tools. In social networks or association networks, for which many network tools were developed, the edges are known with certainty (e.g., social network friends or manuscript co-authors). However, in functional networks inferred from noisy brain activity, the edges are estimated with uncertainty. This uncertainty depends—in complex ways—on the association measure applied and the nature of the data recorded. Different measures to define functional

connectivity may lead to different inferred functional networks and different dynamic communities. How this uncertainty impacts the standard tools of network analysis remains poorly understood. Moreover, these inferred—and uncertain—functional networks change rapidly in time[43,55,56]. Finding the best approaches to infer and characterize dynamic communities from noisy, non-stationary brain signals is a significant challenge. We note that the appropriate choice of plex could change as a function of time. How, why, and the extent to which this would be can be expected to vary with context. However, at a minimum, nontrivial changes in network density over time can be expected to be a factor. In the context of epilepsy, it has been observed that network density can evolve dramatically during a seizure[34]. This might suggest the value of developing an extension of DDPM (as well as related methods like CPM) with adaptively chosen, time-varying plex order.

In this manuscript, we considered a systematic comparison of DPPM with two representative methods, CPM and MMM, as implemented in the literature. However, we note that modifications of these methods—by interchanging characteristics of DPPM, CPM, and/or MMM—would allow a more comprehensive exploration of the benefits and contributions of specific method characteristics to dynamic community detection in general. For example, in DPPM we link networks in time by including an edge from each node to itself, and to all other nodes with which it shares consistent connections (Fig. 1a). This is a generalization of the coupling usually used in MMM; extending MMM to link networks in time as in DPPM would help reveal the impact of this specific method characteristic. In DPPM, the cross-network links proposed have a simple and explicit approach. This approach, for connecting nodes to themselves across time, is implicitly a local smoothing of the network structure itself, with the degree of smoothing connected to the choice of plex k. An alternative approach to link networks in time is a temporal smoothing of network connectivity matrices across adjacent frames. While this approach would link networks in time using all connections, it would also introduce a new smoothing parameter; how to best choose this smoothing parameter automatically is not clear. While we explored a wide range of method variations here, additional modifications may allow CPM and MMM to perform similarly to DPPM (e.g., the choice of null model in MMM, or an alternative procedure to couple networks across layers in CPM). However, such extensions are beyond the scope of the present manuscript.

Understanding the brain's network dynamics remains a fundamental challenge in neuroscience, with opportunities spanning

from genetic networks to social networks[57], and applications spanning health and disease. Here we focus on one component of this challenge: the characterization of dynamic communities in evolving functional connectivity networks, and application to the dynamic networks that emerge during human seizures. While characterization of seizure onset requires inference and analysis of rapidly evolving functional networks, we expect the dynamic community tracking approach developed here will apply to other, multivariate neuronal data sets (e.g., calcium imaging, MEG, multi-neuron recordings). Ultimately, the continued development of statistically principled network analysis tools—combined with advances in data acquisition and computational modeling—is essential to understanding the neural origin, mechanisms, and functions of the brain's dynamic functional connectivity.

## Methods

**DPPM**. The DPPM identifies all dynamic subgraphs over which a $k$-plex of at least order $m$ vertices can be 'walked'. From an algorithmic standpoint, DPPM consists of three subroutines: *Plex*, *StatComm*, and *DynComm*. We briefly describe these subroutines and their implementation here; pseudo-code versions can be found in Supplemental Materials.

Given an input graph on $p$ vertices, say $G = (V,E)$, *Plex* identifies the maximal $k$-plexes, based on ref. [58]. That is, all vertex-induced subgraphs $S_1, S_2,...$ are enumerated such that each $S_j$ is a $k$-plex of size $n_j$ and, if any other vertex is added to $S_j$, it would cease to be a plex. In practice, since a clique is also a $k$-plex and finding a clique is faster, we start by finding all maximal cliques larger than $m$ and then look for $k$-plexes within the remaining vertices that are not parts of any cliques.

Using *Plex*, *StatComm* effectively creates from an input graph $G$ a secondary graph $G^*$, with each vertex corresponding to a maximal $k$-plex $S_j$ in $G$. An edge exists between two vertices $i$ and $j$ in $G^*$ if the number of vertices common to $S_i$ and $S_j$ is at least $m-1$ vertices, indicating that a $k$-plex $S_i$ may be walked to the $k$-plex $S_j$. This step is similar to the CPM that requires a minimum overlap of $m-1$ vertices to aggregate cliques into communities[18]. The connected components in $G^*$, say $C_1^*, C_2^*, ...$, then implicitly represent subsets of vertices/edges in the original input graph $G$ over which a $k$-plex may be walked. These vertices/edges are assigned labels according to their membership in these connected components. A vertex/edge may have multiple labels.

Ultimately, DPPM consists of applying *StatComm* in two passes through a dynamic network $\{G_t\}_{t=1}^N$. Recall that the vertex sets $V_t$ in each graph $G_t$ are assumed to be equal to a common set $V$ of $p$ vertices. In the first pass, *StatComm* is first applied at each time slice $t$ to determine the static communities within each $G_t$. In the second pass, *StatComm* is applied to each of $N-1$ enhanced multi-slice graphs with $2p$ vertices containing pairs of graphs adjacent in time. Each new multi-slice graph, say $G_t^+$ consists of copies of $G_t$ and $G_{t+1}$, wherein a vertex $i$ is labeled $v_t^i$ for its copy from $G_t$ and $v_{t+1}^i$ for its copy from $G_{t+1}$. Next, we enhance this new graph $G_t^+$ by adding an edge between vertices $v_t^i$ and their matured self $v_{t+1}^i$. We also add edges between $(v_t^i, v_{t+1}^j)$ and $(v_{t+1}^i, v_t^j)$ to $G_t^+$ if $(v_t^i, v_t^j)$ is an edge in $G_t$ and $(v_{t+1}^i, v_{t+1}^j)$ is an edge in $G_{t+1}$. This provides a means for the community to walk across time, and visually appears as what we term railroad tracks in $G_t^+$ (see Fig. 1a). Community labels associated with vertices/edges within each of the slices $G_t$ from the first pass of *StatComm* are then propagated forward and backward in time through the resulting augmented dynamic graph $\{G_t^+\}_{t=1}^N$, in an iterative fashion, thus equipping each dynamic community with a unique label.

The computational complexity of DPPM is dominated by the identification of all maximal $k$-plexes in the *Plex* algorithm, which in turn is called in the context of identifying communities across adjacent time points $t$ and $t+1$ using the *StatCom* algorithm. We use a modification of the Bron–Kerbosch (BK) algorithm[58], which is a recursive backtracking procedure for identifying all maximal cliques in an undirected graph consisting of $p$ vertices. Because there are at least as many $k$-plexes as maximal cliques on $m$ vertices, the worst-case complexity for identification of $k$-plexes is at least $O\left(3^{\frac{p}{3}}\right)$. Suppose that $m_t$ and $m_{t+1}$ maximal $k$-plexes are returned by the algorithm on the graphs $G_t$ and $G_{t+1}$, respectively. The static communities at times $t$ and $t+1$ are determined by comparing the sizes of vertex set intersections of all $\binom{m_t}{2} + \binom{m_{t+1}}{2}$ pairs of $k$-plexes within time slices. Similarly, the dynamic communities between times $t$ and $t+1$ are determined by comparing the sizes of vertex set intersections of all $m_t m_{t+1}$ pairs of $k$-plexes across time slices. Let $d_t$ be the number of vertices in the largest maximal $k$-plex in time slice $t$, and $d = \max(d_t, d_{t+1})$. Then each pairwise comparison is $O(d\log d)$ complexity using a standard merge sort. Subsequently, breadth-first-search determines the connected components in $O(m_t + m_{t+1})$ time. In total, letting $m = \max(m_t, m_{t+1})$, identifying communities from time $t$ to $t+1$ has a worst-case complexity of at least $O(d\log dm^2)$. Empirical results indicate that our modified BK

algorithm for identifying maximal $k$-plexes in a graph has time complexity that is output sensitive and performs, experimentally, as sub-exponential.

In practice, several steps are taken to increase computational efficiency. Within *StatComm* the secondary graph $G^*$ is never explicitly constructed. Rather, we simply compute the overlap between all pairs of $k$-plexes and determine community labels accordingly. We use a threshold of $(m-1)$ when considering the vertex overlap between $k$-plexes to build the plex graph and find its connected components. Finally, to avoid having to build a $2p \times 2p$ adjacency matrix from each of the enhanced multi-slice graphs $G_t^+$ we employ a strategy that has been shown (using exhaustive enumeration) to be almost exact for DPPM parameters up to (5,3). Specifically, our heuristic measures the overlap between all pairs of communities found at time steps $t$ and $t+1$ and labels them as the same community if the overlap is greater than $m-1$ vertices. This approach is almost exact in the sense that, when applying DPPM for plexes of size $m$, it reproduces the results of the formal algorithm (i.e., based on the exact rule walking across time slices) except that when encountering an m-cycle, which is matched with its isomorphic motifs (e.g., hourglass to square and vice-versa for size 4). We do not expect this approximation to impact the network communities of interest here for two reasons. First, cycle and isomorphic motifs are unlikely due to transitive connections common in correlation networks[59]. Second, the size of the communities we observe typically exceeds 4 or 5 nodes, so that alternative plexes could stitch the communities in time. We view these trade-offs as acceptable when held against the significant computational improvements the approximation brings when applied to the functional networks of interest here. We find that, in practice, DPPM tends to run in less time than MMM and CPM (Supplementary Fig. 7).

**Existing dynamic community detection methods**. In addition to DPPM, we apply two other dynamic community detection methods already in existence. We implement MMM following the procedure described in ref. [19] and using the MATLAB code (including the GenLouvain function) available at http://netwiki. amath.unc.edu/GenLouvain/GenLouvain, Version 2.1. An additional post-processing step was applied to refine the results: we discard identified communities that are deemed too small (here, size strictly lower than 3) and too short-lived (only found for one time step). We implement CPM following the procedure described in[18] and using in-house MATLAB code available at the repository associated with this paper (see "Code availability").

To estimate the dynamic communities in MMM requires optimization of a quality function with respect to a chosen null model. Different choices exist[60], including approaches that account for functional networks (i.e., networks derived from time series data[30]), which may alter features of the dynamic communities detected (e.g., the number and size of communities). Here we choose the standard null model (the Newman–Girvan null model), which may detect fewer communities of larger size than other approaches[30]. While this choice of null model is common in analysis of neural data[21,23,25,26,41], alternative choices (e.g., Erdős–Rényi) tailored to specific applications may enhance performance. In addition, we note that in practice examination of dynamic functional brain networks requires a comparison of the extracted statistics (such as the module allegiance, flexibility, or laterality) to those expected in a random network null model[30]. This can be done with post-optimization null models, for which different choices exist (e.g., temporal, nodal, and static)[30]. Because we know the true network structure in the simulated data, we did not examine such post-optimization null models here.

While the focus of our work here is specifically motivated by the problem of characterizing the evolution of functional connectivity networks during epileptic seizures, and the details and design choices underlying the proposed DPPM derive directly therefrom, there is of course a large and active literature on dynamic community detection in general. The recent survey by Rosetti and Cazabet[33] offers a concise summary of the literature to date, organized according to a certain taxonomy (see their Fig. 4), with the three main classes of instant-optimal communities discovery, temporal trade-off community discovery, and cross-times communities discovery. In the language of that paper, DPPM is of the third type, specifically of the sub-type evolving memberships, evolving properties. CPM, in contrast, is a method primarily of the first type but arguably with some characteristics of the second type. Finally, MMM is of the third type. The choice of these two algorithms as competitors in our numerical work is therefore representative in the sense that MMM allows for comparison within the same class as DPPM, while CPM allows for comparison across the classes. However, it would be of independent interest—although beyond the scope of the current paper—to evaluate the performance of DPPM broadly, against many methods and on a large compendium of networks.

**Patients and recordings**. The patient analyzed in Fig. 5a–h was a 37-year-old male with medically intractable focal epilepsy who underwent clinically indicated intracranial cortical recordings using grid electrodes for epilepsy monitoring. Clinical electrode implantation, positioning, duration of recordings and medication schedules were based purely on clinical need as judged by an independent team of physicians. The patient was implanted with intracranial subdural grids, strips, and depth electrodes (Adtech Medical Instrument Corporation) for 10 days in a specialized hospital setting and continuous ECOG data were recorded (500 Hz sampling rate). The reference was a strip of electrodes placed outside the dura and

facing the skull at a region remote from the other grid and strip electrodes. One to four electrodes were selected from this reference strip and connected to the reference channel.

Seizure onset times were determined by an experienced encephalographer (S.S.C.) through inspection of the ECOG recordings, referral to the clinical report, and clinical manifestations recorded on video. We selected four seizures to analyze for this patient. Their durations were respectively 75.94 s, 62.79 s, 60.14 s, 133.16 s. We included 100 s before seizure onset in each dataset.

The population of patients analyzed in Fig. 5i–l corresponds to patients P1–P6, P9, P11, P12, P13, P15, and P16 in Table 1 of ref. [40]. We note that, in comparing patients with different surgical outcomes (Fig. 5k, l), we treat multiple seizures from the same patients as independent.

All human subjects were enrolled after informed consent was obtained and approval was granted by local Institutional Review Boards at Massachusetts General Hospital and at Boston University according to National Institutes of Health guidelines.

**Deterministic community detection simulation**. We consider two networks, each of seven nodes. The first network, which exists at time $t$, consists of six edges organized in the adjacency matrix plotted in Fig. 6a. For the second network, which exists at time $t + 1$, we consider the three alternative adjacency matrices plotted in Fig. 6b. The 7-, 8-, and 9-edge networks correspond to Fig. 1c top, middle, and bottom row of plots, respectively. We apply each community detection method to the sequence of two networks: (1) the network at time $t$, and (2) one of the networks selected at time $t + 1$. Because the CPM and MMM do not provide deterministic community assignments, we repeat each method 100 times on the two-network sequence, and indicate the proportion of times each node is assigned to one (of two) communities.

**Robustness to noise simulation**. To illustrate how the three dynamic community detection methods perform when functional network inference is corrupted by false negatives (i.e., missed edges) we perform the following simulation. We begin with a connected nine-node network with adjacency matrix plotted in Fig. 6c. We then simulate a dynamic network for 100 time steps, by removing two edges chosen at random from the adjacency matrix at each time step. We apply each community detection method to the resulting sequence of 100 networks.

**Dynamic network simulations**. We construct simulated network data motivated by multi-electrode brain voltage recordings. To do so we consider a 64-electrode recording, and generate benchmark dynamic network models using the algorithm defined in ref. [61] and available at https://github.com/MultilayerGM/MultilayerGM-MATLAB, Version 1.0.1. For each simulation scenario, we implement a multilayer partition (S) with 64 nodes and either 100 time points (Split, Merge) or 140 time points (Expand, Contract), as described in the following subsections. For concreteness, we assume each layer of the multilayer network represents a functional network inferred for a 1 s interval; i.e., the total duration of a simulated dynamic networks is either 100 s (Split, Merge) or 140 s (Expand, Contract). For each community detection method and for each parameter set investigated for this method, we repeat each simulation scenario 100 times, with different realizations of the multilayer partition S. The parameter sets evaluated for each method are as follows: DPPM $(m,k)$ = (4,1), (4,2), (4,3), (5,1), (5,2) and (5,3); CPM minimum clique size of 3, 4, and 5; MMM all combinations of $\gamma = \{0.01, 0.1, 0.5, 1, 2\}$, $\omega = \{0.01, 0.1, 0.5, 1, 2, 5, 10\}$.

**Simulation scenarios**. In each simulation scenario we define a multilayer partition S to create dynamic communities that evolve in time. To compute the adjacency matrices for each layer, we fix the parameters in ref. [61] as follows: exponent for the power law degree distribution of −2, minimum degree of 3, maximum degree of 20, mixing parameter of 0, maximum number of rejections for a single block of 100.

**Split: splitting of one community into two communities**. In this scenario, we simulate three dynamic communities $c_1$, $c_2$ and $c_3$. Each consists of 15 nodes. We divide the timeline of this scenario into 5 intervals of duration 20 s each, as follows (Fig. 3b):

(1) Initially each node is assigned a random community label (a randomly chosen integer between 1 and 64) for each layer (i.e., 1 s) of the 20 s interval. No dynamic communities exist.
(2) Only community $c_2$ is active; each node receives the same community label.
(3) Communities $c_1$ and $c_3$ become active; each node receives the same community label as $c_2$.
(4) Community $c_2$ becomes inactive (each node receives a random community label), and only communities $c_1$ and $c_3$ remain.
(5) Communities $c_1$ and $c_3$ become inactive; all nodes receive a random community label, as in (1).

**Merge: merging of two communities into one community**. Conceptually, this simulation is the converse of the split scenario. We again simulate three dynamic communities $c_1$, $c_2$ and $c_3$, each consisting of 15 nodes. We divide the timeline of this scenario into 5 intervals of duration 20 s each, as follows (Fig. 3c):

(1) Initially no communities are active; each node is assigned a random community label (a randomly chosen integer between 1 and 64) for each layer (i.e., 1 s) of the 20 s interval.
(2) Communities $c_1$ and $c_3$ become active (each node receives the same community label), without any nodes in common.
(3) Community $c_2$ becomes active, and receives the same community label as $c_1$ and $c_3$.
(4) Communities $c_1$ and $c_3$ become inactive (each node is assigned a random community label), and only $c_2$ remains.
(5) Communities $c_2$ becomes inactive; all nodes receive a random community label, as in (1).

**Expand: one community expands**. In this simulation, we simulate five communities $c_1$, …, $c_5$. We divide the timeline of this scenario into seven equal intervals (20 s duration), such that in each interval we add additional nodes to the community, as follows (Fig. 2):

(1) Initially each node is assigned a random community label (a randomly chosen integer between 1 and 64) for each layer (i.e., 1 s) of the 20 s interval. During this interval, no organized community structure exists.
(2) Community $c_1$ with 15 nodes becomes active; each node receives the same community label.
(3) Community $c_2$ with 10 nodes becomes active; each node receives the same community label as $c_1$.
(4) Community $c_3$ with 10 nodes becomes active, and each node receives the same community label as $c_1$ and $c_2$.
(5,6) We continue to add communities in each interval (i.e., community $c_k$ with 10 nodes becomes active) until all 5 communities have been recruited.
(7) All communities become inactive; all nodes receive a random community label, as in (1).

**Contract: one community contracts**. Conceptually, this simulation is the converse of the scenario Expand. In practice, we implement this scenario, by first performing all of the steps in the Expand simulation. We then reverse the indexing of the time axis for all nodes (i.e., the last instance of activity in the Expand simulation becomes the first instance of activity in the Contract simulation). Doing so results in the following intervals of activity (Fig. 3a):

(1) Initially all communities are inactive; all nodes receive a random community label.
(2) All communities ($c_1$, …, $c_5$) are active.
(3) Community $c_5$ becomes inactive.

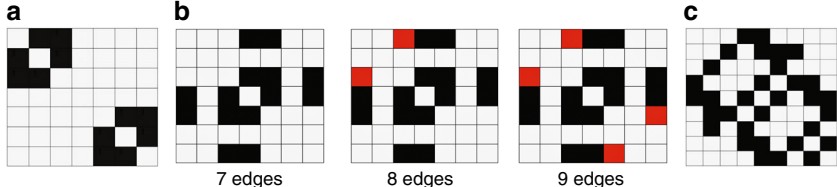

**Fig. 6 Illustration of simple simulated networks.** Adjacency matrices in which black at coordinate $(i, j)$ indicates an edge from node $i$ to node $j$. **a** Seven node network at time $t$. **b** Seven node networks at time $t + 1$, with 7, 8, and 9 edges. Red indicates edges added to the leftmost network. **c** Nine node network.

(4) Community $c_4$ becomes inactive.

(5,6) We continue to remove communities in each interval (i.e., community $c_k$ becomes inactive), until all communities have been removed.

(7) No communities are active.

**Functional connectivity inference**. We apply here a method previously developed in ref. [28] and applied in refs. [56,62,63]. We outline here our specific data analysis approach; a detailed discussion of the measure, including its statistical properties and simulation results, may be found in ref. [28]. Before applying the coupling analysis, we process the time series data in the following way. First, we apply a zero-phase forward and reverse finite impulse response filter of order 1000 to bandpass filter the data between 4 and 50 Hz. Next, we divide the data into 1 s windows with 0.5 s overlap. Finally, we normalize the data from each electrode within the 1 s window to have zero mean and unit variance. We note that we do not re-reference the in vivo data, although the choice of reference can affect the coupling statistics. We use a measure of significant cross correlation, corrected for multiple comparisons, to construct a functional network for each 1 s window[28]. To assess the variability of the cross correlation across lags, we compute the average variance of the cross correlation between all node pairs and all 1 s epochs for the entire dataset; this provides a common measure of variability that we apply to assess the significance of each correlation statistic[28,64].

**Network assessments**. In order to compare how well each dynamic community method tracked each scenario, we determined the similarity between the largest identified community $c_L(t)$ and the true community, $c_T(t)$, at time t. We compute the sensitivity $S^+$ and specificity $S^-$, defined as follows, to quantify this similarity:

$$S^+ = TP/(TP + FN),$$

$$S^- = TN/(TN + FP),$$

where

$$TP = \sum_t |c_T(t) \cap c_L(t)|,$$

$$FN = \sum_t |c_T(t) \cap \overline{c_L(t)}|,$$

$$FP = \sum_t |\overline{c_T(t)} \cap c_L(t)|,$$

$$TN = \sum_t |\overline{c_T(t)} \cap \overline{c_L(t)}|,$$

and the overbar indicates the logical not function.

**Reporting summary**. Further information on research design is available in the Nature Research Reporting Summary linked to this article.

## Data availability

The seizure data that support the findings of this study are available on request from the authors. The data are not publicly available due to them containing information that could compromise research participant privacy/consent.

## Code availability

All analyses and modeling were performed using custom designed algorithms written in MATLAB (MathWorks, Inc). The complete analysis pipeline including the code for simulating the different scenarios is available for re-use and further development at the repository: https://github.com/Eden-Kramer-Lab/dppm. We note that, in the software developed to implement the analysis described here, we have made important aspects of the preprocessing (e.g., type and band of filtering, choice of reference) and network inference (e.g., type of method) easily swappable. Therefore, the interested reader may replace these choices with alternative approaches appropriate for his or her data, and implement the dynamic community tracking algorithm.

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

## Acknowledgements

E.D.K., E.S., L.-E.M., M.A.K, and S.S.C. were supported by the National Institute of Neurological Disorders and Stroke Award R01NS095369. C.J.C. was supported by the National Institute of Neurological Disorders and Stroke Award K23-NS092923. The authors thank Lauren M. Ostrowski and Dan Y. Song for assistance with data collection and processing.

## Author contributions

L.-E.M., M.A.K., and E.K. wrote the manuscript. L.-E.M and M.A.K. analyzed the data and performed the computational modeling. W.V. and E.D.K. developed the DPPM method, and L.N.P. contributed to the computational implementation. L.-E.M., M.A.K., and C.J.C. developed the functional connectivity method. L.-E.M., E.S., and M.A.K. implemented the data analysis pipeline in MATLAB. S.S.C. and C.J.C. developed the application to seizure data. All authors contributed in editing the manuscript.

## Competing Interests

The authors declare no competing interests.
