## [Peer Review File · Nature Communications]

Reviewers' Comments:

Reviewer #1:

Remarks to the Author:

I read this manuscript with much interest, but I have to say that I was left fairly disappointed by the study as currently presented.

The majority of the paper is spent documenting better behavior of "DPPM" compared to "CPM" and "MMM" under some selected artificial scenarios. While I think these scenarios are certainly reasonable to explore, I was left underwhelmed by these studies as justification by themselves for the need for DPPM. The paper simply ignores many other existing methods for community detection. I get it: in many ways your method has a feel similar in spirit to CPM, and MMM is one of the most heavily used methods in this area. Even so, there are other methods you could have used and I think it's important to at least say so. And then even with MMM many of the comparisons here appear to be with selected parameter values with little explanation of why those values were chosen. Figure 1 says $\gamma=1$ and $\omega=1$. Are other values reasonable or perform better? Or were these picked for some reason justifying the best comparison? Figures 2 and 3 says "MMM with parameters (0.01, 1)" but which parameter is which value? And again, why those values? That said, Figure 4 nicely considers different parameter values and adds value by showing apparent trade offs between specificity and sensitivity at different values.

But forgive me for saying this, but up through this point of the paper the whole thing has a very straw-man argument feel to it. You're trying to justify the existence of your method by inventing some test case scenarios and showing better improvement; but I'm not very convinced by what little is here. A more complete series of tests would more clearly delineate behaviors under different parameter values. And what about using the existing multilayer community detection benchmarks from Mason Porter's group? Have you considered using those?

The paper then pivots to application of DPPM to study functional networks from ECOG recordings. I would be far more interested in the paper as a whole if either (a) the point of the paper was about showing how community detection can be used for this application and/or (b) this section demonstrated how other methods perform on this same data. How do CPM and MMM do under different parameter values here? What about any of the other existing methods for community detection in temporal networks? If this is supposed to be a paper introducing and documenting a new method that is essential to study such data, please show us comparisons between DPPM and other methods on this data itself.

As a couple other points about making such comparisons, let me ask about weights and runtimes:

I think it is essential in a network neuroscience context to emphasize which methods can and cannot handle network data with weighted edges, which is a feature of many (not all, but many) different functional and structural network neuroscience data sets. MMM handles weighted edges directly without modification. Weighted versions of CPM exist. What about DPPM? I note the word "weight" doesn't even appear in the current manuscript.

The manuscript says "In practice, several steps are taken to increase computational efficiency." But nothing appears to be said about run times of any of the methods considered here. I think this is important to include. And can anything be said about how the computation might scale with either the numbers of nodes and/or the number of time steps?

I am sorry that I cannot be more positive about the current version of this manuscript. I think there is

something potentially very nice here in the DPPM and its application to data like this. But the current presentation leaves me wanting on the many fronts above. In its current form, I think it's currently far short of the bar needed to be in a Nature journal. I would be very interested in seeing a revision, either to this journal or to another.

Reviewer #2:

Remarks to the Author:

Thank you for inviting me to review this manuscript by Martinet and colleagues, in which the authors introduce a novel Dynamic Plex Percolation Method, which has been designed to robustly track community structure in time-resolved networks (with a particular emphasis on brain imaging data). The method is clearly described, and a number of intuitive examples are provided in order to justify the utility of the approach, particularly in comparison to the recognized 'state of the art' in temporal community detection, along with compelling results from simulated data. The method is then applied to an existing dataset to good effect. Overall, I found the paper to be well reasoned, clearly explained and well justified. I am of the opinion that the work will be of interest to any researchers currently utilizing community detection algorithms on temporal network data. I have only minor comments, that I hope will help the manuscript.

* The authors may wish to comment on whether/how their data could be applied to weighted/signed data, which is commonly used in neuroimaging.

* It would be interesting to know how crucial the appropriate choice of k is for the method to function effectively. That is, does the method break on simulated data with the wrong choice of plex size? And do the authors have any heuristic measures they can suggest that may help researchers unfamiliar with their technique to properly define the value of the plex size in their own datasets? What if the value of the plex changes systematically over time (e.g. as a function of task complexity)? I appreciate that these are difficult problems to solve, but a treatment of the practical issues that users may come across would benefit the manuscript.

Reviewer #3:

Remarks to the Author:

This study describes a new dynamical community detection algorithm. This algorithm is a generalization and extension of the popular clique percolation method. The study illustrates the utility of this algorithm in the context of tracking changes in dynamical communities associated with onset and propagation of seizures.

In principle domain-specific community detection is to be welcomed, but in practice the novelty and improvement in performance of the present algorithm wasn't clear to me from the study's presentation of the simulations and empirical results. I discuss this in more detail by considering the novelty of the algorithm and separately the novelty of the coupling between individual network frames.

Regarding the novelty of the algorithm, it was not clear to me that the present relaxation of cliques to plexes is that much of an original contribution. In general, such relaxation is well known in the sociology literature. In community detection literature it has been discussed by the authors of the clique percolation method (Derenyi et al., 2005), "Although using k -cliques might seem to be a very strict constraint on the community definition, we note that relaxing this constraint (e.g., by allowing

incomplete k -cliques) is practically equivalent to lowering the value of k ". The authors of CPM elsewhere discuss that their two main parameters are k , and the connectivity threshold. Here, in order for the study to claim the superiority of their extension to the original CPM, it needs to carefully explore a range of parameters for both CPM and DPPM, noting that the value of k in general should be lower for CPM cf. DPPM. The results seem to be based primarily on single parameter settings, and it was not clear that these parameters represent the "best fits" for each individual algorithm.

Regarding the coupling between individual network frames, such coupling was also not compared in a systematic manner between algorithms. The described coupling (from a node to itself, and to all other nodes with which it shares consistent connections) is a generalization of the coupling usually used in multilayer modularity maximization, and appears to represent the most novel contribution of the paper. But in order to consider the effect of this coupling systematically, the authors need to evaluate its effects both on their proposed algorithm, and on the multilayer modularity method. I also note that the coupling parameter ω has not been explored. Together, the differences in coupling structure and ω may substantially alter the community structure for both DPPM and MMM.

Moreover, as the authors also state in the manuscript, the coupling between layers is a somewhat ad hoc feature of MMM, and by extension of DPPM. It is not apparent why such coupling should be made from each node to itself (or other nodes) in temporal networks. To me at least, it seems more principled to temporally smooth network connectivity matrices with their adjacent frames, and in this way introduce "coupling" explicitly and between all connections at once (rather than implicitly and only between a small group of connections). What happens to community structure when CPM is ran on such smoothed matrices? In theory, for specific parameter regimes, it should perform similarly to the DPPM.

There are two additional parameters for MMM which could change things: the resolution parameter γ which modulates the number of clusters, and the choice of null model. For the latter, a simpler erdos-renyi null model may prove to be more appropriate for detecting seizure propagation, since it may more effectively model very large communities cf. the configuration model.

In summary, it seems that under a different configuration of parameters relating to clique size, smoothing/coupling, ω , γ , and null model, it should be possible for CPM and/or MMM to perform similarly to the DPPM described by the authors, which implies that the present contribution does not add sufficient novelty to the field of dynamic community detection. I encourage the authors to consider these issues in detail, and to try and illustrate the advantages of their algorithm in a more systematic and exhaustive way.

Other comments:

The "tube detection" built in to clique/plex percolation is known to be a problematic aspect of the algorithm. For example, the algorithm may find a single community in a lattice network.

I wasn't sure why CPM was non-deterministic in this study. Was there a stochastic approximation?

The paragraph on control at the end of the paper does not fit with the rest of the paper.

I think I understand the authors' intent in Figure 1c, but to an unguided observer CPM arguably provides the most intuitive description of the data in this Figure.

Reviewers' comments:

Reviewer #1 (Remarks to the Author):

I read this manuscript with much interest, but I have to say that I was left fairly disappointed by the study as currently presented.

R1.Q1. The majority of the paper is spent documenting better behavior of "DPPM" compared to "CPM" and "MMM" under some selected artificial scenarios. While I think these scenarios are certainly reasonable to explore, I was left underwhelmed by these studies as justification by themselves for the need for DPPM. The paper simply ignores many other existing methods for community detection. I get it: in many ways your method has a feel similar in spirit to CPM, and MMM is one of the most heavily used methods in this area. Even so, there are other methods you could have used and I think it's important to at least say so. And then even with MMM many of the comparisons here appear to be with selected parameter values with little explanation of why those values were chosen. Figure 1 says $\gamma=1$ and $\omega=1$. Are other values reasonable or perform better? Or were these picked for some reason justifying the best comparison? Figures 2 and 3 says "MMM with parameters (0.01, 1)" but which parameter is which value? And again, why those values? That said, Figure 4 nicely considers different parameter values and adds value by showing apparent trade offs between specificity and sensitivity at different values.

R1.A1. In the revised manuscript, we address these important concerns in four ways.

First, we now include in the Supplementary Material the results of Figure 1 with a variety of combinations of MMM parameters (all 35 combinations of $\gamma = \{0.01, 0.1, 0.5, 1, 2\}$ and $\omega = \{0.01, 0.1, 0.5, 1, 2, 5, 10\}$). This figure illustrates why we choose (1,1) as the best performing pair. We now include in the revision the following Supplementary Figure 1:

Supplementary Figure 1. Comparison of MMM computed with different parameter configurations. We apply MMM using all combinations of $\gamma = \{0.01, 0.1, 0.5, 1, 2\}$ and $\omega = \{0.01, 0.1, 0.5, 1, 2, 5, 10\}$ to determine communities across two adjacent time points t and $t+1$. The connected component at time $t+1$ shares increasingly more edges from the top to the bottom row of each figure, as in Figure 1. No combination of gamma and omega shown here

distinguishes the three cases. The color bar indicates the proportion of community membership over 100 repetitions of community detection. The titles $MMM(\gamma, \omega)$ indicate the values of parameters γ and ω .

Second, we now include in the revised Supplementary Material examples of each community detection method applied to each simulation scenario. In doing so, we illustrate how each method performs on the same simulated network dynamics with the following sets of parameters:

DPPM: (4,1), (4,2), (4,3), (5,1), (5,2), and (5,3)

CPM: {3,4,5}

MMM: all 35 combinations of $\gamma = \{0.01, 0.1, 0.5, 1, 2\}$, $\omega = \{0.01, 0.1, 0.5, 1, 2, 5, 10\}$.

We include the four new Supplementary Figures here:

Supplementary Figure 2: Example dynamic community detections applied to a simulation of community expansion. For each method, we consider the following parameter configurations: DPPM (4,1), (4,2), (4,3), (5,1), (5,2) and (5,3); CPM (3,4,5); MMM all combinations of $\gamma = \{0.01, 0.1, 0.5, 1, 2\}$, $\omega = \{0.01, 0.1, 0.5, 1, 2, 5, 10\}$. The true expansion is illustrated in the upper left subfigure. Color indicates community membership. The largest community detected by DPPM is most consistent with the true expansion.

Supplementary Figure 3: Example dynamic community detections applied to a simulation of community contraction. For each method, we consider the parameter configurations listed in the caption of Supplementary Figure 2. The true contraction is illustrated in the upper left subfigure. Color indicates community membership. The largest community detected by DPPM is most consistent with the true contraction.

Supplementary Figure 4: Example dynamic community detections applied to a simulation of a community splitting. For each method, we consider the parameter settings listed in the caption of Supplementary Figure 2. The true splitting is illustrated in the upper left subfigure. Color indicates community membership. The largest community detected by DPPM is most consistent with the true community split.

Supplementary Figure 5: Example dynamic community detections applied to a simulation of a community merging. For each method, we consider the parameter settings listed in the caption of Supplementary Figure 2. The true merging is illustrated in the upper left subfigure. Color indicates community membership. The largest community detected by DPPM is most consistent with the true community merge.

Third, we now make clear in the manuscript text that we choose MMM with parameters $\gamma = 1, \omega = 1$ as an illustrative example for which the MMM performs well across scenarios. When describing Figure 1 in Results, we now state:

“Again, the addition of an edge has little effect on the dynamic communities detected with CPM and MMM. We note that the choice of MMM parameters $\gamma = 1, \omega = 1$ serves as a representative example; other parameter choices perform similarly (see Supplementary Figure 1).”

When describing the first simulation scenario, we now write:

“... We note that here, and in the simulations that follow, we fix the detection method parameters to show representative examples; see Supplementary Figure 2 for examples of each community detection method applied with different parameter settings). ...”

And we now indicate in the captions of Figure 2 and Figure 3 that different parameter choices for each method are shown in Supplementary Figures 2-5.

Forth, we have revised the Discussion to identify other methods that could have been used to track the dynamic communities.

“While the focus of our work here is specifically motivated by the problem of characterizing the evolution of functional connectivity networks during epileptic seizures, and the details and design choices underlying the proposed DPPM derive directly therefrom, there is of course a large and active literature on dynamic community detection in general. The recent survey by Rosetti and Cazabet³⁴ offers a concise summary of the literature to date, organized according to a certain taxonomy (see their Fig. 4), with the three main classes of “instant-optimal communities discovery”, “temporal trade-off community discovery”, and “cross-times communities discovery”. In the language of that paper, DPPM is of the third type, specifically of the sub-type “evolving memberships, evolving properties”. CPM, in contrast, is a method primarily of the first type but arguably with some characteristics of the second type. Finally, MMM is of the third type. The choice of these two algorithms as competitors in our numerical work is therefore representative in the sense that MMM allows for comparison within the same class as DPPM, while CPM allows for comparison across the classes. However, it would be of independent interest – although beyond the scope of the current paper -- to evaluate the performance of DPPM broadly, against many methods and on a large compendium of networks.”

R1.Q2. But forgive me for saying this, but up through this point of the paper the whole thing has a very straw-man argument feel to it. You're trying to justify the existence of your method by inventing some test case scenarios and showing better improvement; but I'm not very convinced by what little is here. A more complete series of tests would more clearly delineate behaviors under different parameter values. And what about using the existing multilayer community detection benchmarks from Mason Porter's group? Have you considered using those?

R1.A2. To address this concern, we have revised the manuscript to utilize the multilayer community detection benchmarks defined in [*Generative benchmark models for mesoscale structure in multilayer networks*, M. Bazzi, L. G. S. Jeub, A. Arenas, S. D. Howison, M. A. Porter. arXiv:1608.06196.] In doing so, we updated the benchmark procedure to generate partitions consistent with the four simulation scenarios of particular interest in our application. We find, consistent with our previous simulation results in the original submission, that DPPM outperforms CPM and MMM.

We have updated the *Methods: Dynamic network simulations* to describe this procedures as follows:

*“We construct simulated **network** data motivated by multi-electrode brain voltage recordings. To do so we consider a 64-electrode recording, and generate **benchmark dynamic network models** using the algorithm defined in ⁵⁹. For each simulation scenario, we implement a **multilayer partition (S)** with 64 nodes and either 100 time points (**Split, Merge**) or 140 time points (**Expand, Contract**), as described in the following subsection. For concreteness, we assume each layer of the multilayer network represents a functional network inferred for a 1 s interval; i.e., the total duration of a simulated dynamic network is either 100 s (**Spilt, Merge**) or 140 s (**Expand, Contract**). For each community detection method and for each parameter set investigated for this method, we repeat each simulation scenario 100 times, with different realizations of **the multilayer partition S**. The parameter sets evaluated for each method **are** as follows: **DPPM** $(m,n) = (4,1), (4,2), (4,3), (5,1), (5,2)$ and $(5,3)$; **CPM** minimum clique size of 3, 4, and 5; **MMM** all combinations of $\gamma = \{0.01,0.1,0.5,1,2\}$, $\omega = \{0.01,0.1,0.5,1,2,5,10\}$.”*

We also updated each section of *Methods: Simulation Scenarios* to describe the use of the benchmark dynamic network models.

Finally, we have updated Figure 2, Figure 3, and Figure 4 in *Results* to display these new simulation results. We note that the new dynamic network simulations do not change the qualitative results presented in the original manuscript.

“Figure 2. In an example of dynamic community expansion, DPPM outperforms two existing methods. (a,b) Illustration of community expansion in nodes (a) and in edges (b). (a) Two-dimensional representation of the nodes on an 8x8 grid at 5 time intervals. Color (blue) indicates when a node becomes recruited to the largest community. (b) Adjacency matrices for the simulated network of 64 nodes at the same 5 time intervals. Color (black) indicates an edge between a node pair. (c) Dynamic community detection results for each method, from left to right: true expansion, DPPM with parameters (4,2), CPM with parameter 4, and MMM with parameters (0.1,0,1). Color indicates community membership. The largest community detected by DPPM is most consistent with the true expansion.”

“Figure 3. In three additional examples of dynamic community evolution, DPPM outperforms two existing methods. Community detection for each method in the case of (a) community contraction, (b) community splitting, and (c) community merging. From left to right in each subfigure: true community evolution, DPPM with parameters (4,2), CPM with parameter 4, and MMM with parameters (0.1, 0.1). Color indicates community membership. In all cases, the largest community detected by DPPM is most consistent with the true community evolution.”

“Figure 4. DPPM performs with higher sensitivity and specificity than two existing methods in the four simulation scenarios. (a) Specificity and sensitivity of DPPM (blue), CPM (red), and MMM (yellow) for 100 simulations of each dynamic community evolution scenario. (a) Community expansion, (b) Community contraction, (c) Community splitting, and (d) Community merging. Each circle indicates the result of one simulation with one parameter configuration (see Methods). (e) Summary results for each community tracking method applied to each simulation scenarios. Bar (lines) indicate the mean (maximum and minimum values) of the sensitivity and specificity for each parameter configuration of each method.”

R1.Q3. The paper then pivots to application of DPPM to study functional networks from ECOG recordings. I would be far more interested in the paper as a whole if either (a) the point of the paper was about showing how community detection can be used for this application and/or (b) this section demonstrated how other methods perform on this same data. How do CPM and MMM do under different parameter values here? What about any of the other existing methods for community detection in temporal networks? If this is supposed to be a paper introducing and documenting a new method that is essential to study such data, please show us comparisons between DPPM and other methods on this data itself.

R1.A3. To emphasize that the point of the paper is to show how community detection can be used for this application, we have updated the manuscript to illustrate in more detail the application of DPPM to the human seizure data. We now extend the DPPM analysis to include a population of patients and seizures (12 patients, 38 seizures). We find that, consistent with the example patient and seizures described in the original manuscript, that: (i) the largest community size increases during seizure compared to pre-seizure, and (ii) the lifespan of the largest community is longer during seizure compared to pre-seizure. Moreover, we now compare these dynamic network features with a clinical result. We first made the hypothesis that patients with a worse surgical outcome would have more fractured, longer-lasting dynamic communities during seizure, which are less susceptible to a targeted intervention (in this cases, resective surgery). We show that both the number of communities and the duration of the longest community increase in patients with worse surgical outcomes. In this way, we now illustrate how the proposed community detection method can be used for our specific application.

We have updated Figure 5, and the text describing that figure, as follows:

“Figure 5. Application of DPPM reveals new characteristics of dynamic communities before and after human seizure onset. (a) Top panel: Voltage time series recorded at 9 electrodes to illustrate pre-ictal and ictal voltage dynamics. Middle panel: Example recruitment of a large community at seizure onset. Before seizure onset ($t < 0$ s) small communities appear briefly. After seizure onset ($t > 0$ s) a large dynamic community appears (red) that persists for over 30 s. Bottom panel: Temporal evolution of the size of the seizure onset community (red). Nearly all nodes participate in the seizure onset community. (b) Example expansion of the seizure onset community over the brain surface. Each circle denotes an electrode on the 8x8 electrode grid, and red (black) indicates electrodes recruited (not yet recruited) into the dynamic community. (c,d) Community lifespan (c) and maximum community size (d) for all pre-seizure communities (gray histograms) and the seizure onset community observed for each of the 4 seizures of this

patient (4 arrows, the red arrow indicates the community shown in (a)). **(e,f)** Node loyalty averaged over the 4 seizures for (e) all pre-seizure communities and (f) the seizure onset community. In both panels, warm (cool) colors indicate nodes that participate in communities for longer (shorter) times. The black circles indicate a subset of electrodes that have high node loyalty both before and during early seizure. **(g)** Median recruitment order to the seizure onset community for 4 seizures. Warm (cool) colors indicate electrodes recruited earlier (later) into the seizure onset community. **(h)** Mean recruitment time to large amplitude ictal oscillations observed for the same patient, as reported in ⁶³. Warm (cool) colors indicate electrodes recruited earlier (later) into ictal spread. **(i, j)** Histograms of the maximum community size (j) and the lifespan of this maximum community during the pre-seizure (blue) and seizure (red) intervals from each patient and seizure. The maximal community tends to be larger and of longer duration during seizure. **(k, l)** The mean number of communities (k) and the longest community duration (l) during seizure for patients with good (Engel 1,2) and poor (Engel 3,4) surgical outcomes. Each asterisk indicates an individual seizure, and the red square the population mean. Worse surgical outcomes exhibit more communities with longer maximal duration during seizure.”

We have updated the corresponding Results to read:

“To explore further these initial observations we repeat the analysis for a population of patients and seizures (12 patients, 38 seizures). We find, consistent with the example patient and seizures, that: (i) the largest community size increases during seizure compared to pre-seizure (Figure 5i), and (ii) the lifespan of the largest community is longer during seizure compared to pre-seizure (Figure 5j). We then explored the hypothesis that patients with a worse surgical outcome would have more fractured, longer-lasting dynamic communities during seizure, which are less susceptible to a targeted intervention (in this cases, resective surgery). We find that both the number of communities (Figure 5k) and the duration of the longest community (Figure 5l) increase in patients with worse surgical outcomes (higher Engel scores; see Methods).

Overall, we conclude from these observations that application of DPPM to ECOG data recorded from patients with epilepsy provides new insights into the expansion of a large dynamic community at seizure onset. We propose that larger communities of longer duration emerge during seizure, and that patients with fewer, shorter duration communities during seizure have improved surgical outcomes.”

As a couple other points about making such comparisons, let me ask about weights and runtimes:

R1.Q4. I think it is essential in a network neuroscience context to emphasize which methods can and cannot handle network data with weighted edges, which is a feature of many (not all, but many) different functional and structural network neuroscience data sets. MMM handles weighted edges directly without modification. Weighted versions of CPM exist. What about DPPM? I note the word "weight" doesn't even appear in the current manuscript.

R1.A4. To address this important concern, we have updated the Discussion of the revised manuscript as follows:

“Third, DPPM is designed for binary networks that are often inferred from multi-electrode brain recordings, but in neuroscience it is frequently desirable to analyze weighted networks. CPM has been extended to weighted networks⁴⁹, where in order to percolate cliques are now required to be of sufficient intensity (i.e., the geometric mean of the link weights in a clique⁵⁰ must exceed a given threshold). An extension of DPPM to weighted networks should be similarly feasible, although somewhat less immediate for two reasons. First, there can be a multiplicity of plexes among a given set of nodes, and thus there is flexibility in representation. One approach might be to adopt the recently-introduced notion of a maximal edge-weighted plex⁵¹. Second, it is necessary to equip the edges between the same nodes at different network time points with an appropriate notion of an edge weight. Depending on the manner of network construction, there may be multiple ways in which to do so. A full and careful exploration of these possibilities is beyond the scope of the present paper. We note that MMM handles weighted edges seamlessly and without modification.”

R1.Q5. The manuscript says "In practice, several steps are taken to increase computational efficiency." But nothing appears to be said about run times of any of the methods considered here. I think this is important to include. And can anything be said about how the computation might scale with either the numbers of nodes and/or the number of time steps?

R1.A5. To address this concern, we now include the new Supplementary Figure 6 to illustrate the run times for each method applied to the four simulation scenarios:

“Supplementary Figure 6: DPPM tends to run in less time than MMM and CPM. Run times for each community detection method applied to each simulation scenario. Each simulation scenario (row) shows the means (bar heights) and standard deviations (vertical black lines) of the run times calculated for each method (columns) for 100 realizations of the simulated dynamics networks. The bottom subfigure lists the community detection method and the parameter configuration. The parameter configurations include DPPM: $\{(4,2), (5,3)\}$; MMM: all 35 combinations of $\gamma = \{0.01, 0.1, 0.5, 1, 2\}$ and $\omega = \{0.01, 0.1, 0.5, 1, 2, 5, 10\}$; CPM: $\{3,4,5,6\}$.”

In addition, we have updated Method to include the following theoretical analysis:

“The computational complexity of our DPPM method is dominated by the identification of all maximal k -plexes in the Plex algorithm, which in turn is called in the context of identifying communities across adjacent time points t and $t+1$ using the StatCom algorithm. We use a modification of the Bron-Kerbosch (BK) algorithm⁵⁶, which is a recursive backtracking procedure for identifying all maximal cliques in an undirected graph consisting of p vertices. Because there

are at least as many k -plexes as maximal cliques on m vertices, the worst-case complexity for identification of k -plexes is at least $O\left(3^{\frac{p}{3}}\right)$. Suppose that m_t and m_{t+1} maximal k -plexes are returned by the algorithm on the graphs G_t and G_{t+1} , respectively. The static communities at times t and $t+1$ are determined by comparing the sizes of vertex set intersections of all $\binom{m_t}{2} + \binom{m_{t+1}}{2}$ pairs of k -plexes within time slices. Similarly, the dynamic communities between times t and $t+1$ are determined by comparing the sizes of vertex set intersections of all $m_t m_{t+1}$ pairs of k -plexes across time slices. Let d_t be the number of vertices in the largest maximal k -plex in time slice t , and $d = \max(d_t, d_{t+1})$. Then each pairwise comparison is $O(d \log d)$ complexity using a standard merge sort. Subsequently, breadth-first-search determines the connected components in $O(m_t + m_{t+1})$ time. In total, letting $m = \max(m_t, m_{t+1})$, identifying communities from time t to $t+1$ has a worst case complexity of at least $O(d \log d m^2)$. Empirical results indicate that our modified BK algorithm for identifying maximal k -plexes in a graph has time complexity that is output sensitive and performs, experimentally, as sub-exponential."

R1.Q6. I am sorry that I cannot be more positive about the current version of this manuscript. I think there is something potentially very nice here in the DPPM and its application to data like this. But the current presentation leaves me wanting on the many fronts above. In its current form, I think it's currently far short of the bar needed to be in a Nature journal. I would be very interested in seeing a revision, either to this journal or to another.

R1.A6. We thank the Reviewer for the instructive comments, which we believe have substantially improved the revised manuscript.

Reviewer #2 (Remarks to the Author):

R2.Q0. Thank you for inviting me to review this manuscript by Martinet and colleagues, in which the authors introduce a novel Dynamic Plex Percolation Method, which has been designed to robustly track community structure in time-resolved networks (with a particular emphasis on brain imaging data). The method is clearly described, and a number of intuitive examples are provided in order to justify the utility of the approach, particularly in comparison to the recognized 'state of the art' in temporal community detection, along with compelling results from simulated data. The method is then applied to an existing dataset to good effect. Overall, I found the paper to be well reasoned, clearly explained and well justified. I am of the opinion that the work will be of interest to any researchers currently utilizing community detection algorithms on temporal network data. I have only minor comments, that I hope will help the manuscript.

R2.A0. We are glad the Reviewer found the work interesting, and we thank the Reviewer for the positive feedback.

R2.Q1. * The authors may wish to comment on whether/how their data could be applied to weighted/signed data, which is commonly used in neuroimaging.

R2.A1. To address this comment, we now include a discussion of how the method could be extended to apply to weighted networks. Please see R1.A4 for our detailed response.

R2.Q2. * It would be interesting to know how crucial the appropriate choice of k is for the method to function effectively. That is, does the method break on simulated data with the wrong choice of plex size? And do the authors have any heuristic measures they can suggest that may help researchers unfamiliar with their technique to properly define the value of the plex size in their own datasets? What if the value of the plex changes systematically over time (e.g. as a function of task complexity)? I appreciate that these are difficult problems to solve, but a treatment of the practical issues that users may come across would benefit the manuscript.

R2.A2. We address these important concerns in three ways. First, we now show the results for the simulations for different parameter choices in each dynamic community detection method. For the DPPM method, we see in general that using plexes with $k=1$ (i.e., of the form $(m,1)$) generally yields inferior results. Please see R1.A2 - and specifically Figure 4e,f - for details.

Second, we motivate why this result might be expected, through the use of network motifs. The authors of [32] hypothesize motifs in neuronal functional networks to be important to the brain architectures of highly evolved species. The authors of [33] concluded that higher order connection patterns, which they termed themes, i.e. an aggregation of motifs, were significantly discovered in neurological functional networks. We show in Supplementary Table 1, for a variety of non-trivial motifs on 5 or fewer vertices, the extent to which these may be represented by one of, or an interleaving of, a plex of order $m=3, 4, \text{ or } 5$. We see that plexes with $k=1$ have the poorest ability to represent these network motifs.

(n, k)	Subgraphs														
(3, 1)															
(4, 1)															
(5, 1)															
(4, 2)															
(5, 3)															
(5, 2)															

Supplementary Table 1: Network motifs detectable via plexes. We display all subgraphs on $m=3, 4, \text{ or } 5$ vertices which can be discovered by order $k=1, 2, \text{ and } 3$ plexes. The top row represents the motifs which we seek to discover and which will form the themes as an aggregation thereof. The next six rows represent the order (m,k) plex which can discover such a motif in the respective column.

Third, we now address in the revised Discussion the issue plex choice changing as a function of time.

“... We note that the appropriate choice of plex could change as a function of time. How, why, and the extent to which this can be expected to vary with context. However, at a minimum, nontrivial changes in network density over time can be expected to be a factor. In the context of epilepsy, it has been observed that network density can evolve dramatically during a seizure³⁵. This might suggest the value of developing an extension of DDPM (as well as related methods like CPM) with adaptively chosen, time-varying plex order. However, such an extension is beyond the scope of the present manuscript. ...”

Reviewer #3 (Remarks to the Author):

This study describes a new dynamical community detection algorithm. This algorithm is a generalization and extension of the popular clique percolation method. The study illustrates the utility of this algorithm in the context of tracking changes in dynamical communities associated with onset and propagation of seizures.

In principle domain-specific community detection is to be welcomed, but in practice the novelty and improvement in performance of the present algorithm wasn't clear to me from the study's presentation of the simulations and empirical results. I discuss this in more detail by considering the novelty of the algorithm and separately the novelty of the coupling between individual network frames.

R3.Q1. Regarding the novelty of the algorithm, it was not clear to me that the present relaxation of cliques to plexes is that much of an original contribution. In general, such relaxation is well known in the sociology literature. In community detection literature it has been discussed by the authors of the clique percolation method (Derenyi et al., 2005), “Although using k -cliques might seem to be a very strict constraint on the community definition, we note that relaxing this constraint (e.g., by allowing incomplete k -cliques) is practically equivalent to lowering the value of k ”. The authors of CPM elsewhere discuss that their two main parameters are k , and the connectivity threshold. Here, in order for the study to claim the superiority of their extension to the original CPM, it needs to carefully explore a range of parameters for both CPM and DPPM, noting that the value of k in general should be lower for CPM cf. DPPM. The results seem to be based primarily on single parameter settings, and it was not clear that these parameters represent the “best fits” for each individual algorithm.

R3.A1. To address this specific concern, we now explore a range of parameters for both CPM and DPPM in the four simulation scenarios. More specifically, we now compute DPPM and CPM with different values of k ; we choose $k=\{3,4,5\}$ for CPM and $k=\{4,5\}$ for DPPM. Please see example results in the new Supplementary Figures 1-5 (included in R1.A1 above) and a more complete exploration of parameter settings in the new Figure 4 (included in R1.A2 above).

We now specifically compare DPPM and CPM, with a lower value of k for CPM. We find across the four simulation scenarios that $k=3$ maximizes the sensitivity and specificity for CPM (see the new Figure 4). While DPPM and CPM perform with similar specificity across parameter settings, DPPM(4,2), DPPM(4,3), and DPPM(5,3) perform with higher sensitivity than CPM($k=3$). We now discuss this important point in the revised manuscript as follows:

“Based on the results from these four simulation categories (Figure 4), we conclude that only DPPM detects the dynamic community behavior with high sensitivity and specificity in all scenarios considered. We find that no fixed parameter setting for MMM performs with both high sensitivity and high specificity across all simulation scenarios. While DPPM and CPM perform with similar specificity across parameter settings, DPPM(4,2), DPPM(4,3), and DPPM(5,3) perform with higher sensitivity than CPM($k=3$).”

R3.Q2. Regarding the coupling between individual network frames, such coupling was also not compared in a systematic manner between algorithms. The described coupling (from a node to itself, and to all other nodes with which it shares consistent connections) is a generalization of the coupling usually used in multilayer modularity maximization, and appears to represent the most novel contribution of the paper. But in order to consider the effect of this coupling systematically, the authors need to evaluate its effects both on their proposed algorithm, and on the multilayer modularity method. I also note that the coupling parameter ω has not been explored. Together, the differences in coupling structure and ω may substantially alter the community structure for both DPPM and MMM.

R3.A2. In the revised manuscript, we now explore the impact of changing the parameter ω on MMM. These results are summarized in the revised Figure 4 (included in R1.A2 above). We also now include example results for different values of parameter γ in the new Supplementary Figures 1-5 (included in R1.A1 above).

Inclusion of these additional results is consistent with our original goal of a systematic comparison of our proposed DPPM method with two top, representative competitors from the literature. That is, with two competitors in their *existing* forms. The reviewer makes several intriguing suggestions (here and below) for comparison of DPPM to additional methods that currently do not exist but which could in principle be developed by interchanging various characteristics of DPPM, CPM, and/or MMM. We agree with the reviewer that development of these various additional new algorithms could potentially be interesting and (if successful) would certainly allow for a more comprehensive and systematic understanding of the relative benefits and contributions of these characteristics to dynamic community detection in general. But we respectfully submit that such a study is well beyond the scope of this paper and would

be better suited either for a handful of (likely small) papers or, for example, in a survey or synthesis paper in the general literature on community detection. Nevertheless, for completeness we have tried, where possible below, to provide some feedback on these suggestions for various novel algorithms.

R3.Q3. Moreover, as the authors also state in the manuscript, the coupling between layers is a somewhat ad hoc feature of MMM, and by extension of DPPM. It is not apparent why such coupling should be made from each node to itself (or other nodes) in temporal networks. To me at least, it seems more principled to temporally smooth network connectivity matrices with their adjacent frames, and in this way introduce “coupling” explicitly and between all connections at once (rather than implicitly and only between a small group of connections). What happens to community structure when CPM is ran on such smoothed matrices? In theory, for specific parameter regimes, it should perform similarly to the DPPM.

R3.A3. What happens to CPM when we smooth is an interesting question (but, as we say above, beyond the scope, as it amounts to an additional (semi)novel algorithm). Importantly, note that if you were to smooth, you then typically have to introduce another parameter, i.e., the smoothing parameter. So now, in principle, we are trying to see if we can find some setting of that parameter where we match or beat DPPM with CPM. But we’d need a method to choose the smoothing parameter automatically and how best to do so is not clear. The seminal work of Wasserman and colleagues [Zhou, S., Lafferty, J., & Wasserman, L. (2010). Time varying undirected graphs. *Machine Learning*, 80(2-3), 295-319] in the context of smoothing stationary dynamic networks for covariance estimation is illustrative of the subtleties that quickly arise (and networks in our context have the added complication of being nonstationary). Whereas, with the cross-network links (i.e., the coupling), we have a simple and explicit approach. And this approach, for connecting nodes to themselves across time, is implicitly a local smoothing of the network structure itself, with the degree of smoothing connected to the choice of plex k . So while it does indeed have an ad hoc flavor to it, we would argue that it is nevertheless getting at the goal of smoothing raised by the reviewer and in a way that is simple and direct to tune.

R3.Q4. There are two additional parameters for MMM which could change things: the resolution parameter gamma which modulates the number of clusters, and the choice of null model. For the latter, a simpler erdos-renyi null model may prove to be more appropriate for detecting seizure propagation, since it may more effectively model very large communities cf. the configuration model.

R3.A4. In the revised manuscript, we now explore the impact of changing the parameter gamma on MMM. These results are summarized in the revised Figure 4 (included in in R1.A2 above). We also now include example results for different values of parameter gamma in the new Supplementary Figures 1-5 (included in R1.A1 above).

Modification of the choice of null model in the MMM methodology – beyond that already adopted and implemented – is an interesting idea. The reviewer suggests exploring the Erdos-

Renyi null model but, of course, there are a variety of other options that could be explored, a thorough pursuit of which would make for an interesting future manuscript. Here our goal is to compare the performance of DPPM to two top, existing methods as currently implemented and understood in the literature. Our results show that DPPM has a clear advantage in the specific application setting motivating the work.

R3.Q5. In summary, it seems that under a different configuration of parameters relating to clique size, smoothing/coupling, omega, gamma, and null model, it should be possible for CPM and/or MMM to perform similarly to the DPPM described by the authors, which implies that the present contribution does not add sufficient novelty to the field of dynamic community detection. I encourage the authors to consider these issues in detail, and to try and illustrate the advantages of their algorithm in a more systematic and exhaustive way.

R3.A5. In response to the Reviewer's comments, we have updated the manuscript to describe in detail the impact of clique size, omega, and gamma in the four simulation scenarios. These are what we would characterize as the 'standard' tuning parameters associated with the published forms of the CPM and MMM algorithms (and amenable to manipulation by the average user through the existing software implementations available to the broader community). As explained above, however, we respectfully submit that a systematic and exhaustive comparison of the type the reviewer encourages us to pursue is – realistically – developing additional (semi)novel variants of these algorithms for dynamic community detection. And, as such, a comprehensive effort of this nature is better suited for a survey or synthesis manuscript. We now mention this important point in the revised manuscript in the Discussion as follows:

“Additional modifications may allow CPM and MMM to perform similarly to DPPM (e.g., the choice of null model in MMM, or an alternative procedure to couple networks across layers in CPM). However, such extensions are beyond the scope of the present manuscript.”

Other comments:

R3.Q6. The “tube detection” built in to clique/plex percolation is known to be a problematic aspect of the algorithm. For example, the algorithm may find a single community in a lattice network.

R3.A6. This is an interesting statement. Could the reviewer provide a reference for additional detail? It would seem that such detail will matter here in assessing the extent to which the outcome of a “tube detection” algorithm makes sense or not. For example, presumably a static network is not meant by “lattice” here, but we are not sure specifically what kind of (evolving?) lattice the reviewer has in mind.

R3.Q7. I wasn't sure why CPM was non-deterministic in this study. Was there a stochastic approximation?

R3.A7. We now note that, in some cases, the CPM stitching algorithm must arbitrarily decide which communities propagate in time. For example, in Figure 1c, the label that continues from time t to $t + 1$ is chosen arbitrarily (i.e., by a fair coin flip) from the existing community labels at time t , as prescribed in [Palla et al, Uncovering the overlapping community structure of complex networks in nature and society. Nature, 435:814 -- 818, 2005]. Therefore, the community that continues from time t to $t + 1$ is chosen arbitrarily.

“... We note that, unlike the deterministic result of DPPM, the community label at time $t+1$ for CPM and MMM is not unique. In this example, the community label for CPM at time $t+1$ is chosen arbitrarily (i.e., by a fair coin flip) from the existing community labels at time t ³⁸. In the second example, we add an additional edge to the single connected component ...”

R3.Q8. The paragraph on control at the end of the paper does not fit with the rest of the paper.

R3.A8. To address this comment, we now omit this paragraph from the revised manuscript.

R3.Q9. I think I understand the authors' intent in Figure 1c, but to an unguided observer CPM arguably provides the most intuitive description of the data in this Figure.

R3.A9. To address this concern, we have updated the manuscript to include an illustration of the preferred result for each edge case. We have revised Figure 1 as follows:

“Figure 1. Illustration of DPPM principles and effectiveness. (a) Schematic of the bridges used to walk plexes within dynamic communities across time. Blue edges represent inferred connectivity, while red edges connecting the same (solid) or adjacent (dotted) vertices facilitate movement. (b) Illustrative example showing how a simple community (in orange) is tracked by DPPM across time in a manner allowing for both coalescence and fragmentation. This community at first grows at $t + 1$, and then fragments at $t + 2$. Another community present at t (purple) perishes at $t + 1$. (c) Comparison of DPPM (with 2-plex of size $n=3$, 2nd column), CPM (with 1-plex of size $n=3$, 3rd

column), and MMM (with $\gamma = 1, \omega = 1$, 4th column) in determining communities across two adjacent time points t and $t + 1$. The connected component at time $t + 1$ shares increasingly more edges components at time t from the top to the bottom row of plots. In the preferred results, the dynamic communities (1st column) depend on the number of edges shared from time t to $t+1$. Whereas DPPM treats these as three distinct scenarios, neither CPM nor (effectively) MMM distinguish the three cases. In CPM and MMM, the colorbar indicates the proportion of community membership over 100 repetitions of community detection. **(d)** Example dynamic community tracking in the presence of missing edges. Dynamic community membership for ten example sequential time index networks computed using DPPM (with 2-plex of size $n=4$), CPM (with 1-plex of size $n=3$), and MMM (with $\gamma = 1, \omega = 1$). While DPPM detects a single dynamic community in time, the other two methods do not.”

Reviewers' Comments:

Reviewer #1:

Remarks to the Author:

I am pleased by the serious and thorough manner in which the authors addressed by constructive criticisms. I support publication of the current version of the manuscript.

Reviewer #2:

Remarks to the Author:

The authors have adequately addressed my concerns.

Reviewer #3:

Remarks to the Author:

The authors have responded thoroughly to some of my suggestions, and have argued that my other suggestions lie beyond the scope of the paper because they constitute the development of "additional (semi)novel variants of these algorithms".

I think the word (semi)novel is key here, insofar it is applied to the paper more generally. To summarize, the paper combines simulations, and an empirical example, to demonstrate an improved performance by an extension of an existing algorithm. Does such an extension constitute a novel algorithm, or is it an incremental advance? It is difficult to give a clear answer to this question. On the one hand, from an engineering perspective, this makes no difference as long as the algorithm performs better, and thus facilitates translation. But on the other hand, from a scientific perspective, this makes a lot of difference insofar as these distinctions can either increase understanding (of dynamic communities and their detection), or conversely increase confusion through a proliferation of similar-yet-different approaches.

In this context, I think it is reasonable to conclude that the present paper makes progress in engineering, through better tools, but possibly hinders progress in science, through more confusion. In defense of the paper, such conceptual confusion is all too common in the literature of community detection, and has affected some of the most prominent papers in the field including, for that matter, the modularity paper itself (Newman, 2006).

But at the same time, it is also not optimal that the paper has not sufficiently emphasized the links between different community algorithms and their parameter settings, especially since the authors have discussed the importance of these links in response to my comments. One sentence hidden at the end of the discussion ("Additional modifications...") is not enough to make readers who may otherwise be unfamiliar with these algorithms understand the connections between them. Instead, it would be much more preferable to devote at least one paragraph in the methods section to a discussion of these connections, including the effects of links cliques/plexes, coupling and null models on the nature and performance of individual algorithms. In the simplest case, such paragraph could cover the discussion of R3.A2-5, that already touches on many of these points.

Other comments

* For consistency, it makes sense to also add the exploration of CPM and DPMM parameters to Supplementary Figure 1, in the same way that it was added to Supplementary Figures 2-5.

* The large parameter space in the new supplementary figures makes it difficult to easily evaluate the performance of distinct parameter regimes. It would help here to add heatmaps of partition similarity (e.g. normalized mutual information with ground truth) that would quantitatively illustrate this performance at a glance.

* The motivation of the algorithm with motifs is unfortunately unsupported by the available evidence. Motifs, at least at the macroscale, probably represent a byproduct of local brain connectivity (see the series of papers by Markov, Kennedy and collaborators). More generally, it is very difficult to substantively link the present algorithm to some causal motif-driven computation inferred from correlational data.

REVIEWERS' COMMENTS:

Reviewer #1 (Remarks to the Author):

I am pleased by the serious and thorough manner in which the authors addressed by constructive criticisms. I support publication of the current version of the manuscript.

Reviewer #2 (Remarks to the Author):

The authors have adequately addressed my concerns.

Reviewer #3 (Remarks to the Author):

R3.Q1 The authors have responded thoroughly to some of my suggestions, and have argued that my other suggestions lie beyond the scope of the paper because they constitute the development of “additional (semi)novel variants of these algorithms”.

I think the word (semi)novel is key here, insofar it is applied to the paper more generally. To summarize, the paper combines simulations, and an empirical example, to demonstrate an improved performance by an extension of an existing algorithm. Does such an extension constitute a novel algorithm, or is it an incremental advance? It is difficult to give a clear answer to this question. On the one hand, from an engineering perspective, this makes no difference as long as the algorithm performs better, and thus facilitates translation. But on the other hand, from a scientific perspective, this makes a lot of difference insofar as these distinctions can either increase understanding (of dynamic communities and their detection), or conversely increase confusion through a proliferation of similar-yet-different approaches.

In this context, I think it is reasonable to conclude that the present paper makes progress in engineering, through better tools, but possibly hinders progress in science, through more confusion. In defense of the paper, such conceptual confusion is all too common in the literature of community detection, and has affected some of the most prominent papers in the field including, for that matter, the modularity paper itself (Newman, 2006).

But at the same time, it is also not optimal that the paper has not sufficiently emphasized the links between different community algorithms and their parameter settings, especially since the authors have discussed the importance of these links in response to my comments. One sentence hidden at the end of the discussion (“Additional modifications...”) is not enough to make readers who may otherwise be unfamiliar with these algorithms understand the connections between them. Instead, it would be much more preferable to devote at least one paragraph in the methods section to a discussion of these connections, including the effects of links cliques/plexes, coupling and null models on the nature and performance of individual algorithms. In the simplest case, such paragraph could cover the discussion of R3.A2-5, that already touches on many of these points.

R3.A1. We appreciate these observations, and have revised the manuscript to include these important points in the Discussion as follows:

“In this work, we considered a systematic comparison of DPPM with two representative methods, CPM and MM, as implemented in the literature. However, we note that modifications of these methods – by interchanging characteristics of DPPM, CPM, and/or MMM – would allow a more comprehensive exploration of the benefits and contributions of specific method characteristics to dynamic community detection in general. For example, in DPPM we link networks in time by including an edge from each node to itself, and to all other nodes with which it shares consistent connections (Figure 1a). This is a generalization of the coupling usually used in MMM; extending MMM to link networks in time as in DPPM would help reveal the impact of this specific method characteristic. In DPPM, the cross-network links proposed have a simple and explicit approach. This approach, for connecting nodes to themselves across time, is implicitly a local smoothing of the network structure itself, with the degree of smoothing connected to the choice of plex k . An alternative approach to link networks in time is a temporal smoothing of network connectivity matrices across adjacent frames. While this approach would link networks in time using all connections, it would also introduce a new smoothing parameter; how to best choose this smoothing parameter automatically is not clear. While we explored a wide range of method variations here, additional modifications may allow CPM and MMM to perform similarly to DPPM (e.g., the choice of null model in MMM, or an alternative procedure to couple networks across layers in CPM). However, such extensions are beyond the scope of the present manuscript.”

In addition, we now mention the choice of an alternative null model in the Discussion as follows:

“To estimate the dynamic communities in MMM requires optimization of a quality function with respect to a chosen null model. Different choices exist⁵⁸, including approaches that account for functional networks (i.e., networks derived from time series data³⁰), which may alter features of the dynamic communities detected (e.g., the number and size of communities). Here we choose the standard null model (the Newman-Girvan null model), which may detect fewer communities of larger size than other approaches³⁰. While this choice of null model is common in analysis of neural data^{21,23,25,26,42}, alternative choices (e.g., Erdős-Rényi) tailored to specific applications may enhance performance. In addition, we note that in practice examination of dynamic functional brain networks requires a comparison of the extracted statistics (such as the module allegiance, flexibility, or laterality) to those expected in a random network null model³⁰. This can be done with post-optimization null models, for which different choices exist (e.g., temporal, nodal, and static)³⁰. Because we know the true network structure in the simulated data, we did not examine such post-optimization null models here.”

Other comments

R3.Q2. * For consistency, it makes sense to also add the exploration of CPM and DPMM parameters to Supplementary Figure 1, in the same way that it was added to Supplementary Figures 2-5.

R3.A2. In Figure 1c, we consider CPM and DPPM for $n=3$. Exploration of additional CPM and DPPM parameters would require increasing n . *A priori* these choices of n are too large to capture the relevant community structure in these small, toy graphs; we note that only communities of size $n=3$ exist at time t in Figure 1c. The purpose in Figure 1c is illustration, which we achieve by zooming in on a tiny, local patch of what we would envision to be a larger network. But without the surrounding network, these larger values of n are not as meaningful.

R3.Q3. * The large parameter space in the new supplementary figures makes it difficult to easily evaluate the performance of distinct parameter regimes. It would help here to add

heatmaps of partition similarity (e.g. normalized mutual information with ground truth) that would quantitatively illustrate this performance at a glance.

R3.A3. Thank you for the suggestion. In the main manuscript, we compare the inferred dynamic communities with the ground truth by computing the sensitivity and specificity (as in Figure 4). To use these same measures here, we compute the F_1 score:

$$F1 = 2 \frac{PPV \times sensitivity}{PPV + sensitivity}$$

which combines the positive predictive value (PPV) and sensitivity into a single descriptive statistic that is easy to visualize. We now include in Supplementary Figure 6 heatmaps of the F_1 score to illustrate the performance of each simulation at a glance.

R3.Q4. * **The motivation of the algorithm with motifs is unfortunately unsupported by the available evidence. Motifs, at least at the macroscale, probably represent a byproduct of local brain connectivity (see the series of papers by Markov, Kennedy and collaborators). More generally, it is very difficult to substantively link the present algorithm to some causal motif-driven computation inferred from correlational data.**

R3.A4. We thank the reviewer for noting this concern. To address this, we have updated the manuscript to provide a more general motivation for the use of motifs. We have updated the manuscript as follows:

“The proposed method extracts dynamic communities based on the explicit notion of time-evolving aggregations of smaller motifs, which have been proposed as building blocks ~~of brain function characteristic of different types of networks~~^{31,32,34} ~~and as important evolutionary outcomes in the neuronal architectures of highly evolved species~~³². The new method of dynamic community detection – the dynamic plex percolation method (DPPM) – connects static communities within a time-slice to aggregations of a variety of common motifs in a natural and flexible manner, defines dynamic communities across time through an explicit notion of temporal progression of these motifs, and is demonstrably robust to edge noise.”

“The central role played by plexes in the DPPM framework derives from the fact that plexes are network elements consistent with motifs, the building blocks of larger network structures^{31,32,33,37} ~~and particularly of brain networks~~³⁴ (Supplementary Table 1). Our dynamic communities, explicitly defined as aggregations of such building blocks, are thus consistent in spirit with ~~current~~ notions of network (sub)structure in ~~the brain network~~ sciences.”

“We developed approximations to reduce computational time in two cases - (4,2) and (5,3) - which are consistent with the size of structural and functional motifs proposed as ~~the common~~ network building blocks^{31,32} ~~of the brain~~⁴⁶ ~~and with motifs observed to occur frequently in biological and technological systems~~³². DPPM successfully aggregates these common motifs into communities. ~~We note that the appropriate interpretation of network motifs in neuroscience remains a point of open discussion. While motifs have been proposed as network building blocks of the brain~~⁴⁶, motifs may instead represent a byproduct of local brain connectivity^{47,48}. ~~The connection of DPPM with motifs through the central role of plexes may facilitate more explicit study of this issue going forward, in the specific context of dynamic community detection.”~~